# Prediction of source contributions to urban background PM$_{10}$ concentrations in European cities: a case study for an episode in December 2016 using EMEP/MSC-W rv4.15 - Part.2 The city contribution

Matthieu Pommier[1,*]

1 Norwegian Meteorological Institute, Oslo, Norway
* now at Ricardo Energy and Environment, Harwell, Oxfordshire, UK.

Contact: matthieu.pommier@ricardo.com

**Abstract.**

Despite the progress made in the latest decades, air pollution is still the primary environmental cause of premature death in Europe. The urban population risks more likely to suffer to pollution related to high concentrations of air pollutants such as in particulate matter smaller than 10 µm (PM$_{10}$). Since the composition of these particulates varies with space and time, the understanding of the origin is essential to determine the most efficient control strategies.

A source contribution calculation allows to provide such information and thus to determine the geographical location of the sources (e.g. city or country) responsible for the air pollution episodes. In this study, the calculations provided by the regional EMEP/MSC-W rv4.15 model in a forecast mode, with a 0.25° longitude × 0.125° latitude resolution, and based on a scenario approach, have been explored. To do so, the work has focused on event occurring between 01 and 09 December 2016. This source contribution calculation aims at quantifying over 34 European cities, the "city" contribution of these PM$_{10}$, i.e. from the city itself, on an hourly basis. Since the methodology used in the model is based on reduced anthropogenic emissions, compared to a reference run, the choice of the percentage in the reductions has been tested by using three different values (5%, 15% and 50%). The definition of the "city" contribution, and thus the definition of the area defining the cities is also an important parameter. The impact of the definition of these urban areas, for the studied cities, was investigated (i.e. 1 model grid cell, 9 grid cells and the grid cells covering the definition given by the Global Administrative Area - GADM).

Using a 15% reduction in the emission and the use of larger cities for our source contribution calculation (e.g. 9 grid cells and GADM), help to reduce the non-linearity in the concentration changes. This non-linearity is observed in the mismatch between the total concentration and the sum of the concentrations from different calculated sources. When this non-linearity is observed, it impacts the $NO_3^-$, $NH_4^+$ and H$_2$O concentrations. However, the mean non-linearity represents only less than 2% of the total modelled PM$_{10}$ calculated by the system.

During the studied episode, it was found that 20% of the surface predicted PM$_{10}$ had been from the "city", essentially composed of primary components. 60% of the hourly PM$_{10}$ concentrations predicted by the model came from the countries in the regional domain, and they were essentially composed of $NO_3^-$ (by ~35 %). The two other secondary inorganic aerosols are also

important components of this "Rest of Europe" contribution, since $SO_4^{2-}$ and $NH_4^+$ represent together almost 30% of this

contribution. The rest of the $PM_{10}$ was mainly due to natural sources. It was also shown that the Central European cities were

mainly impacted by the surrounding countries while the cities located a little away from the rest of the other European countries

(e.g. Oslo and Lisbon) had larger "city" contribution. The usefulness of the forecasting tool has also been illustrated with an

example in Paris, since the system has been able to predict the primary sources of a local polluted event on 01-02 December

2016 as documented by local authorities.






## 1. Introduction.

Air pollution is progressing up in the list of policy priorities for most the industrialized countries. However, even in Europe, progress still have to be made to reduce the levels of pollutant in the air. As shown by the European Environment Agency (EEA), most people living in European cities are exposed to poor air quality (EEA report 2017). The European Court Auditors (ECA) also stipulated that air pollution is the biggest environmental risk to health in the European Union, with about 400,000 people who die each year prematurely due to excessive air pollutants (ECA, Special report 2018). They concluded that the European countries still not sufficiently protect their citizens' health. This shows that additional efforts need to be done at local and regional scales to improve the air quality.

One of this pollutant, the particulate matter smaller than 10 µm ($PM_{10}$), is related to premature mortality at high exposure. The World Health Organization (WHO) has established a short-term exposure $PM_{10}$ guideline value of 50 µg/m$^3$ daily mean that should not be exceeded in order to ensure healthy conditions (WHO, 2005). The WHO has also established a stricter guideline value for the annual average at 20 µg/m$^3$. In Europe, even if the air quality has been improved during the last decade, 13% of the EU-28 urban population was exposed to $PM_{10}$ levels above the daily limit value and approximately 42 % was exposed to concentrations exceeding the annual WHO guideline value in 2016 (EEA report 2018).

These $PM_{10}$ can be emitted locally or transported on long distance. Most of the episodes occur in winter (e.g. EMEP Status Report 1/2018). Indeed, in wintertime, these episodes are often caused by a combination of stagnant air conditions and enhanced use of wood burning for residential heating during cold weather situations. The agriculture and the road traffic have also a large impact even if these two sources are known to usually contribute to $PM_{10}$ pollution in spring (e.g. EEA report 2018; EMEP Status Report 1/2018). More generally, the origin of the $PM_{10}$ can be anthropogenic such as the car traffic and agriculture as mentioned, the industry and the fuel combustion; and also natural such as the desert dust which can largely affect cities as Barcelona (e.g. Perez et al., 2012; Titos et al., 2017), sea salt which has a large impact over the coastal cities (e.g. Hama et al., 2018) and emitted by the forest fires (e.g. Slezakova et al., 2013; Turquety et al., 2020). The $PM_{10}$ are composed of primary components such as organic matter (OM), elemental carbon (EC), dust, sea salt, and other compounds. The $PM_{10}$ are also composed of secondary components compounds formed by chemical reactions in the atmosphere from gas-phase precursors, such as nitrate ($NO_3^-$), ammonium ($NH_4^+$), sulphate ($SO_4^{2-}$), and a large range of secondary organic aerosol (SOA) compounds. These secondary aerosols can represent a large fraction of the $PM_{10}$ composition in European cities (e.g. Querol et al., 2004; Amato et al., 2016; Redington et al., 2016, Diapouli et al., 2017). These $PM_{10}$ are essentially removed from the atmosphere by wet deposition, even if dry deposition over different types of surface may have an important role (e.g. Mitchell et al., 2010; Fuzzi et al., 2015). The variety of sources for these different components highlight the importance to estimate properly the source contributions in air quality modelling.

To provide information to identify the sources of the polluted events over different European cities, a forecasting source apportionment product has been developed within the Copernicus Atmosphere Monitoring Service (CAMS). The predictions

are calculated for 4 days and are available on the website https://policy.atmosphere.copernicus.eu/SourceContribution.php.

The calculations are provided for the surface $PM_{10}$ and its different components over European cities. The predictions are done as a complement to the country source contribution calculations, providing information on the countries responsible of the same polluted events. These country contributions are described in a companion paper (Pommier et al., 2020). The calculations, presented in this study, separate the city contribution from external contributions. Thus, by combining the information from the country contribution given in the companion paper and the city contribution presented hereafter, the system allows

providing information on long-range transport in the European cities and the pollution coming from the urban area. These contributions might be important to determine short term air pollution control measures, which can remain difficult to assess by local authorities.

During the last decade, a few methodologies have been applied to estimate the city contribution to surface $PM_{10}$ concentrations over the European cities through a modelling approach. For example, the SHERPA tool (Thunis et al., 2016), the TM5-FASST

source-receptor model (Crippa et al.,2017) and the GAINS integrated assessment model (Kiesewetter et al., 2015), to cite a few, assume a linear relationship between concentration and emission changes. While the SHERPA tool bases its estimation on model scenarios from other regional models (EMEP/MSC-W model and CHIMERE), the GAINS model combines past monitoring data with bottom-up emission modelling and a simplified atmospheric chemistry and dispersion calculation. The TM5-FASST model is based on a set of emission perturbation experiments as done in our work. However, the emission

perturbation experiment, or also named the scenario approach, may cause non-linearity, i.e. the concentration changes resulting from these perturbations over different sources are not necessarily equivalent to the sum of the individual contribution from all these sources (e.g. Clappier et al., 2017). This shows that the impact of the non-linearity should be analysed for the estimation of the source contribution.

None of the cited studies have provided daily or hourly predictions of city contributions, whereas information is needed to

explain the origin of limit value exceedances in cities throughout Europe. Thus, the objective of this study is to present the near-real time calculation of the urban background contribution predicted by the EMEP/MSC-W model on hourly resolution for each capital of the 28 European Union countries plus Barcelona, Bern, Oslo, Reykjavik, Rotterdam and Zurich. For the simplicity of the reading, the EMEP/MSC-W model is hereafter referred to as EMEP model. This study has been focused on an event occurring in Europe between the 01 and 09 December as described in Pommier al. (2020). This event was the first

event listed from the beginning of the development of system. Pommier et al. (2020) have already shown this event was mainly related to emissions of the Domestic country, i.e. coming from the country corresponding to the studied city such as France for Paris, while the influence of other countries was mainly characterized by a large fraction of $NO_3^-$. However, the contribution from the city, included in this Domestic Country contribution, was not estimated in this companion paper.

For the calculation of this "City" contribution, the definition of the city area is a critical parameter. For this reason, the domain

defining the studied cities was investigated. It is worth noting, the definition uses a relatively coarse resolution (at least 0.25°

longitude × 0.125° latitude) which is representative of the background concentration, and is comparable to the definition of the city domain used in previous studies such as in Thunis et al. (2016) who used an area of $35 \times 35$ km$^2$ or in Skyllakou et al. (2014) who used a radius of 50 km from the city center. Thus, 1 model grid cell (0.25° longitude × 0.125° latitude), 9 grid cells and the grid cells covering the definition given by the Global Administrative Area - GADM) have been used as also done in Pommier al. (2020). Pommier et al. (2020) found by using a larger domain defining the cities helps to limit the impact of the chemical non-linearity in the predictions. In this work, the "City" contribution corresponds to the averaged concentration over a studied city. It is worth noting in our definition of the "city" contribution, there is no distinction between the urban background and the rural background which both may impact the concentration of the pollutant over a city as explained in Thunis et al. (2018).

Section 2 provides a short introduction of the model set-up, i.e. a description of the model and of the experiment. Section 3 details the methodology used in the source contribution (SC) calculation. Section 4 explains the information calculated by the SC during the episode. Section 5 describes the portion of the "City" contribution over the European cities during the episode. Finally, the conclusions are given in Section 6.

## 2. The model set-up

### 2.1. The EMEP model

The EMEP model is an Eulerian model described in detail in Simpson et al. (2012). Initially, the model has been aimed at European simulations, but global scale modelling has been possible for many years (e.g. Wild et al., 2012) and applications over other regions have already been done, such as in India (Pommier et al., 2018) and in China (Brasseur et al., 2019). The EMEP model version rv4.15 has been used here in the forecast mode. The version rv4.15 has been described in Simpson et al. (2017) and references cited therein. The main updates since the version presented in Simpson et al. (2012) and used in this work, concern a new calculation of aerosol surface area (now based upon the semi-empirical scheme of Gerber, 1985), a revised parameterizations of $N_2O_5$ hydrolysis on aerosols, an additional gas-aerosol loss processes for $O_3$, nitric acid (HNO$_3$) and hydroperoxy radical (HO$_2$), a new scheme for ship NO$_x$ emissions, a new calculated natural marine emissions of dimethyl sulfide (DMS), the use of a new land-cover (used to calculate biogenic VOC emissions and the dry deposition) and an update in the source function for sea salt production to account for whitecap area fractions, following the work of Callaghan et al. (2008) (Simpson et al., 2016 and 2017).

The chemical scheme couples the sulphur and nitrogen chemistry to the photochemistry using about 140 reactions between 70 species. The chemical mechanism is based on the "EMEP scheme" described in Simpson et al. (2012) and references therein. The biogenic emissions of isoprene and monoterpene are calculated in the model by emission factors as a function of temperature and solar radiation (Simpson et al., 2012).

In the EMEP model, PM emissions are split into EC, OM (here assumed inert) and the remainder, for both fine and coarse PM. The OM emissions are further divided into fossil-fuel and wood-burning compounds for each source sector. As in Bergström et al. (2012), the OM/OC ratios of emissions by mass are assumed to be 1.3 for fossil-fuel sources and 1.7 for wood-burning sources. The model also calculates windblown dust emissions from soil erosion. The sea salt generation is based on two source functions, those of Monahan et al. (1986) and Mårtensson et al. (2003) as described in Tsyro et al. (2011). Secondary aerosol consists of inorganic sulphate, nitrate and ammonium, and SOA; the latter is generated from both anthropogenic and biogenic emissions, using the 'VBS' scheme detailed in Bergström et al (2012) and Simpson et al. (2012). The EMEP model uses the MARS equilibrium module of Binkowski and Shankar (1995) to calculate the partitioning between gas and fine-mode aerosol phase in the system of $SO_4^{2-}$-HNO$_3$-$NO_3^-$-NH$_3$-$NH_4^+$. This module also calculates the mass of aerosol water (Simpson et al., 2012). This calculated mass of water is added to dry PM$_{10}$ masses when being compared with measured concentrations.

The main loss process for particles is wet-deposition, and the model calculates in-cloud and sub-cloud scavenging of gases and particles as detailed in Simpson et al. (2012). Wet scavenging is treated with simple scavenging ratios, taking into account in-cloud and sub-cloud processes.

In the EMEP model, the 3D precipitation is needed and an estimation of this 3D precipitation can be calculated by the model if this parameter is missing in the meteorological fields. This estimate is derived from large scale precipitation and convective precipitation accumulated at surface. The height of the precipitation is derived from the cloud water. Then, it is defined as the highest altitude above the lowest level, where the cloud water is larger than a threshold taken as $1.0 \times 10^{-7}$ kg water per kg air. Precipitation is only defined in areas where surface precipitation occurs. The intensity of the precipitation is assumed constant over all heights where they are non-zero and is set equal to surface precipitation intensity.

Meteorological data are normally required at 3-hourly intervals for the EMEP model. The EMEP model has systems for deriving parameters when missing or can do without some meteorological fields such as the 3D precipitation explained above. Table S1 summarises the meteorological fields used in the EMEP model. Vertically, the fields are interpolated onto the 20 EMEP σ levels.

Gas and particle species are also removed from the atmosphere by dry deposition. This dry deposition parameterization follows standard resistance-formulations, accounting for diffusion, impaction, interception, and sedimentation.

## 2.2. The experiment

The studied episode occurred from 01 to 09 December 2016 and the forecasts provided by the EMEP model cover Europe (30°N-76°N, 30°W-45°E) (Pommier et al., 2020). An initial spin-up of 10 days was conducted. The model provides four-day air quality forecasts, and the predicted fields have been used to initialise successive four-day forecasts. These predictions were driven by forecasted meteorological fields at 12UTC from the previous day, with a 3-hour resolution, calculated by the Integrated Forecasting System (IFS) of ECMWF. These forecasted meteorological fields correspond to the fields which were

used in the online production for these dates and used in the companion paper (Pommier et al., 2020). The ECMWF forecasts do not include 3D precipitation, which is needed by the EMEP model as mentioned in Section 2.1. Therefore, a 3D precipitation estimate is derived from IFS surface variables (large scale and convective precipitations). A comparison of the calculations by using other meteorological fields, such as reanalysis has not been provided in this work.

The boundary conditions (BCs) at 00UTC of the current day from the atmospheric Composition module (C-IFS) have been used. These BCs are specified for ozone ($O_3$), carbon monoxide (CO), nitrogen oxides (NO and $NO_2$), methane ($CH_4$), $HNO_3$, peroxy-acetyl nitrate (PAN), $SO_2$, ISOP, ethane ($C_2H_6$), some VOCs, sea salt, Saharan dust and $SO_4$.

The TNO-MACC emission dataset for 2011 on $0.25° \times 0.125°$ (longitude-latitude) resolution (Kuenen et al., 2014, see https://atmosphere.copernicus.eu/sites/default/files/repository/MACCIII_FinalReport.pdf) has been used and the forest fire emissions are from GFASv1.2 inventory (Kaiser et al., 2012) as done in the companion paper and at the beginning of the development of the product. It is worth noting the use of a more recent CAMS emission product (CAMS-REG, Granier et al., 2019) has not been addressed in this work.

Since this study aims quantifying the city contribution from each city, the effect of the choice of the city domain has been tested. The city edge has been defined by 1 grid cell (i.e. $0.25°$ lon $\times 0.125°$ lat, corresponding to the emissions data set resolution), 9 grid cells and the all the grid cells covering the administrative area provided by the database of Global Administrative Areas (GADM, https://gadm.org/data.html). This latter is the more precise definition in terms of buildup area, however it may represent a large region as shown in Fig. 1a, such as Riga. It is also clear with Fig 1b that the 9 grid cells domain corresponds to an extension of the 1 grid cell domain; and the area using the GADM definition may differ from the two other definitions as over Ljubljana and in Switzerland. The advantage to have a city domain defined by the 1 grid cell or 9 grid cells, is to have a similar domain for all cities used for the comparison. By using the grid cells based on GADM definition, the size of the cities differs according to the administrative extension of each city.

The natural contributions are defined in this study as the sum of the contributions from sea salt, dust and forest fires.

## 3. Methodology of the EMEP source contribution calculation

### 3.1 Scenario approach: Emission reductions

The SC calculation follows the methodology uses in the country SC calculations (Pommier et al., 2020). The methodology is a scenario approach and consists in estimating the concentration changes by performing and subtracting two simulations. In our case, we have compared a reference run, where all the anthropogenic emissions are included, with a perturbation run, where the emissions over a specific source are reduced. These perturbation runs which correspond to the simulations where the emissions from every considered source region (e.g. a city) are reduced by 15%. As explained in Wind et al. (2004), a reduction of 15% is sufficient to give a clear signal in the concentration changes. It also gives a negligible effect from non-linearity in the chemistry. In the companion paper, it was shown that the non-linearity, related to the emissions reduction used,

represented less than 2% of the total concentrations over each city (Pommier et al., 2020). As performed in this companion study, the effect of the non-linearity, related to the percentage used in the perturbated simulations, has been estimated in this work.

The perturbations are done for anthropogenic emissions of CO, $SO_x$, $NO_x$, $NH_3$, non-methane volatile organic compounds (NMVOC) and primary particulate matter (PPM). As mentioned in Section 2.1, these PPM are distinguished in the EMEP model for two size of aerosols, fine aerosols and coarse aerosols. Note that, except on $NH_3$, the main source regions of these anthropogenic emissions such as $NO_x$ and CO are located over the main urban areas as shown in Fig. S1. For computational efficiency, all anthropogenic emissions in the perturbation runs have been reduced simultaneously. It is worth noting that the non-linearity related to this simultaneous reduction in emissions have not been addressed in this work for computational reason. Indeed, reducing the emissions simultaneously or separately may lead to a different result in the concentrations (e.g. Thunis et al., 2015).

The perturbation runs have been performed for each capital of the 28 European Union countries plus Barcelona, Bern, Oslo, Reykjavik, Rotterdam and Zurich. These simulations over these selected cities, in comparison with the reference run, give the contribution for each city. For convenience, these city SC simulations were gathered by pair, such as Tallinn and Athens. It means that the pair of cities has their emissions reduced simultaneously. These pairs of cities have been chosen to do not impact on each other. In total, there are 17 pair runs.

In addition, there is also a run where the external influence defined as "Rest of Europe" has been performed. This run presents reduced emissions over all the countries within the regional domain. Since this additional perturbated run also includes the cities, this "Rest of Europe" contribution has been calculated by subtracting the "City" contribution. The calculated concentration of the pollutant integrated over the studied city, corresponds to the difference between the integrated concentration from the reference run and the integrated concentration of the perturbation run, scaled by 15%. By differentiating over the studied area, the concentration from the perturbed run with the concentration provided by the reference run, we have an estimation of the influence of the source (i.e. city). By scaling with the reduction used, it gives the estimated concentration related to the source.

The remaining $PM_{10}$ which are neither included in the "City" contribution nor in the "Rest of Europe" contribution are listed in the "Extra sources" contribution which is mainly represented by the BCs and natural sources (sea salt, forest fires and dust). Thus, all these simulations are a complementary information of the country contributions presented in Pommier al. (2020). Indeed, in the country contribution calculations provided in Pommier et al. (2020), there is the "Domestic country" which represents the country corresponding to the studied city (e.g. Spain for Barcelona). Another contributor in the country SC is "30 European countries". In the country SC, the contributions for 31 countries are calculated which include the 28 EU countries, Iceland, Norway and Switzerland, and this "30 European countries" combines all these contributors and excludes the "Domestic country".

## 3.2 Limitation of the methodology: the chemical non-linearity

As explained previously, the calculated concentrations based on a scenario approach, may be impacted by non-linearity. The calculated concentrations due to a reduced emission depend on the atmospheric composition already presents. The total $PM_{10}$ over the receptor should be theoretically identical to the sum of the $PM_{10}$ originated from the different sources, but due to this non-linearity, this is not always the case and it might have few differences between the total $PM_{10}$ and the sum from the various sources.

To ensure the robustness of the methodology, as done in Pommier et al. (2020), the 15% perturbation has been tested and values of 5% and 50% in the perturbation runs were also used. By using these three different perturbations, the total number of simulations performed for this study is equal to 495: 17 pairs city × 9 dates (from 01 to 09 Dec) × 3 perturbations (5%, 15%, 50%) + 9 rest of Europe (one per day) × 3 perturbations (5%, 15%, 50%) + 9 reference runs (one per day).

To reduce simultaneously or separately the emissions may result different non-linearities. However, this difference on the non-linearity, in response to these emission changes has not been quantified for computational reason.

## 4. Information provided by the source contribution calculations during the episode

### 4.1 Evaluation of the predicted concentrations

It is worth noting for this episode in December 2016, the predictions in $PM_{10}$ concentrations of the EMEP model over the cities were compared in the companion paper to predictions provided by another chemistry transport model, LOTOS-EUROS (Manders et al., 2017), and airbase measurements (see https://www.eea.europa.eu/data-and-maps/data/airbase-the-european-air-quality-database-8#tab-data-by-country). It has been shown both models behaved similarly, and it was noticed when the EMEP model predicted larger $PM_{10}$ concentrations it was due to larger secondary inorganic aerosols concentrations than in LOTOS-EUROS. At the opposite, when LOTOS-EUROS predicted more $PM_{10}$, it was due to larger natural components than the EMEP model. The comparison with the $PM_{10}$ measurements highlighted better agreement with the rural stations, which can be located in our city areas due to the coarse definition of these areas, than with urban stations. Pommier et al. (2020) found a maximum correlation coefficient of 0.78 with the rural sites and 0.5 with the urban sites. The EMEP model also underestimates the $PM_{10}$ concentrations by 36 % on average by using the urban sites, and overestimates the concentrations by 6 % compared to the measurements of the rural stations. The differences seen with the measurements may also be related to uncertainties in the regional emission inventory as regards to local situations and in the meteorological fields since forecasted meteorological fields have been used, but the impacts of the choice of the emission inventory and of the meteorological fields have not been addressed in this work. However, the meteorological conditions as used in the EMEP model were well represented over most of the cities, as shown in the comparison with the measurements of the NOAA Integrated Surface Hourly Data Base (https://www.ncdc.noaa.gov/isd) in Table S2. For example, by gathering all cities, the wind speed at 10 m has a correlation coefficient of 0.84 and a normalised mean bias of 8.08%, the relative humidity at 2 m has a correlation coefficient

of 0.59 and a normalised mean bias of -2.38%, and the temperature at 2 m has a correlation coefficient of 0.95 and a normalised mean bias of -0.13%. It is worth noting in some cities, the wind speed is overestimated, which may cause an overestimation in the dispersion of the pollutants.

### 4.2 Origin of the $PM_{10}$

290    In December 2016, a PM episode occurred across North-Western Europe, as a consequence of a high-pressure system Europe (see http://policy.atmosphere.copernicus.eu/reports/CAMSReportDec2016-episode.pdf). December 2016 was one of the warmer Decembers that Europe has ever known. For example, the United Kingdom reported its eighth warmest December in a series dating to 1910. In Norway, December temperature was 4.6°C above its 1961–1990 national average, making this one of the 10 warmest Decembers in the country's 117-year period of record. In a same time, December 2016 was drier than the

295    normal, except in Norway. France was record dry, with average precipitation totals only 20 percent of its 1991–2010 average, breaking the previous record low of December 2015, and Austria had the driest December, where precipitation records date back to 1851 (NOAA, Global Climate Report for December 2016).

High concentrations were measured and predicted over Paris (Fig. 2); and on December 6th and 7th, concentrations at some measurement stations in France, Belgium, the Netherlands, Germany and Poland, exceeded the daily WHO limit value of 50

300    µg/m³ (Pommier et al., 2020). Some examples of these large concentrations for different dates are shown in Fig S2. Even if the larger peaks are missed by the model, the predictions were able to capture the variability of the $PM_{10}$ concentrations over the cities at different dates.

Figure 2 shows the "City" contribution, and the "Rest of Europe" contribution have also been estimated, gathering the concentrations from all the European countries included in the regional domain. There are also the "Extra sources" which

305    gather essentially the natural sources and the BCs. As a complementary information, the reader is invited to compare with the Figure 1 in the companion paper, presenting the country contributors for the same time-series. By combining the information from both timeseries, it is clear that the contribution from France in Paris was largely influenced by the city itself and not only by the rest of the country.

Figure 3 presents the mean composition for the "City", "Rest of Europe and "Extra sources" $PM_{10}$ contributions for all cities,

310    for all 4-day predictions (from 01-04 Dec to 09-12 Dec) and split into negative and positive concentrations. The sum of each contribution should correspond to the total $PM_{10}$ calculated by the reference run, but some differences can appear. By splitting the $PM_{10}$ concentrations for each contribution based on their sign, the negative $PM_{10}$ concentrations help to reveal the species impacted by the non-linearity and explaining the differences seen with the total $PM_{10}$ concentrations calculated by the reference run. On the other hand, the positive concentrations provide the information on the overall composition for each contribution.

315    The figure shows the main contributors to the "City" $PM_{10}$ are the primary components, i.e. EC, POM and rest PPM (which corresponds to the remainder of coarse and fine PPM) as showed by the positive concentrations (Fig. 3a). These three primary

components represent between 70% and 80% of the predicted "City" $PM_{10}$. This large influence of primary components in the "City" contribution is predicted for all cities and for each day as shown in Figs. S3-S6.

The value of the mean $PM_{10}$ concentration depends on the city definition and so on the average of the concentrations over different size of city (1 grid cell, 9 grid cells, GADM). The mean $PM_{10}$ concentration in a smaller area is larger, since the 1 grid cell is the closest grid to the emission source and so the mean concentration is less dispersed than over a larger area.

The "Rest of Europe" $PM_{10}$ is mainly influenced by $NO_3^-$ (by ~35%) (Fig. 3b). This agrees with the result given in the companion paper by the EMEP country SC, showing that the $PM_{10}$ coming from 30 European countries have been composed of 38% of $NO_3^-$ (Pommier et al., 2020). The other secondary inorganic aerosols represent ~13% for $SO_4^{2-}$ and ~14% for $NH_4^+$ in this "Rest of Europe" contribution while the rest PPM remains an important component with ~ 12%, as also shown in Fig. S5. The large influence of the secondary inorganic aerosols and especially $NO_3^-$ is calculated for the whole period (Figures S7 – S10).

Overall, the city SC shows only 20% of the surface $PM_{10}$ calculated over the selected cities during this episode have been from the "City" due to the primary components and another 20% have been from the "Extra sources" mainly composed of natural sources (~60-70%). 60% of the contributions to the surface $PM_{10}$ have been coming from the "Rest of Europe", essentially $NO_3^-$ (by ~35%). The two other secondary inorganic aerosols represent another important part of this "Rest of Europe" contribution, since the $SO_4^{2-}$ and $NH_4^+$ together represent almost 30%.

It shows that the main contributor of the $PM_{10}$ during the episode was caused by the long-range transport. Since there is a low contribution from cities, and the country SC showed that the main contributor was the "domestic country", that means the "Rest of Europe" contribution is mainly composed of this "domestic country". In other words, that means this episode was mainly influenced by the "Domestic" country and not by the cities.

### 4.3 Impact of the non-linearity for each contribution

In Figure 3, the non-linearity has been highlighted by the negligible negative contributions calculated for the "City" and "Rest of Europe" contributions and small negative contributions predicted in "Extra sources". As explained in Section 3.1., the non-linearity and thus, these negative $PM_{10}$ are a result of the assumed linearity in the chemistry to full reduction by using a perturbation factor (5%, 15% or 50%). This impacts the $NO_3^-$, $NH_4^+$ and $H_2O$ (aerosol water content) concentrations as shown in Fig. 3, which is a consequence of gas-aerosol partitioning of the species.

These species are linked through chemical reactions. $NH_3$ may react with nitric acid ($HNO_3$) to form ammonium nitrate ($NH_4NO_3$). This is an equilibrium reaction, and thus the transition from solid to gaseous phase depend on relative humidity (e.g. Wang et al., 2020), explaining why the $NO_3^-$, $NH_4^+$ and $H_2O$ concentrations are linked. In addition to this, the effect of the change in emissions depends on the atmospheric composition already present. This means that the results based on a scenario approach as in our calculation will depend on the chemical regime. For example, an amount of $NO_x$ emitted over a

source can result in a certain $NH_4NO_3$ concentration in the city. When $NO_x$ is emitted in excess, i.e. within a $NH_3$ limited regime, a $NO_x$ emission reduction will have a small effect at the receptor point. Thus, the combination of $NO_x$ and $NH_3$ chemical regimes within different source regions may lead at the end to a mismatch between the sum of the contributions and the total $PM_{10}$, resulting to these negative concentrations. However, this non-linear effect only leads to negative concentrations less than 0.2 µg.m$^{-3}$ (0.8%) of the mean $PM_{10}$ concentrations.

The impact of the percentage used in the perturbation runs and the size of the city edges have no significant impact in the amount of negative "Extra sources" $PM_{10}$ concentrations and the impact of both parameters is very small on the "city" and "Rest of Europe" concentrations (Fig. 3). As in the country SC, the use of larger grids reduces the amount of the negative $PM_{10}$ concentrations and reduces globally the impact of the non-linearity. The 15% factor also reduces the negative non-linearity in the "City" concentrations (e.g. $H_2O$ for the 9 grids and GADM runs).

Similarly to the methodology used in the country source apportionment (Pommier et al. 2020), we have compared the $PM_{10}$ concentrations calculated by using the different percentages in the perturbation runs over the same city edges (Fig. 4). By comparing the three estimates from the perturbation runs to the total concentration for each contribution, this gives an estimation of the impact of the non-linearity for each contribution. In theory, the three perturbated runs should provide the same hourly $PM_{10}$ concentration than the reference run. The non-linearity has been calculated for each hourly contribution (which can be positive or negative as shown in Fig. 3), as the standard deviation of the hourly contribution obtained by the three reduced emissions scenarios, and weighted by the hourly mean of the total concentration by following the equation (1):

$$NONLIN_{Contrib} = \frac{\sqrt{\frac{\sum_{i=1}^{n}\left(Ccontrib_i - \overline{Ccontrib}\right)^2}{n}}}{Ctot} \times 100\% \tag{1}$$

n corresponds to the number of perturbations used (n=3), Ccontrib is the hourly $PM_{10}$ concentration for a specific contribution ("City" or "Rest of Europe" or "Extra sources") and Ctot is the hourly $PM_{10}$ concentration.

The mean non-linearity due to the "City" contribution represents in maximum 0.3% of the total $PM_{10}$, and it represents in maximum 1.7% from the "Rest of Europe" and the "Extra sources" as shown in Figure 4. It is worth reminding the "Extra sources" contribution is calculated by subtracting the total $PM_{10}$ concentrations to the two other contributions. Thus, the non-linearity from the "Extra sources" depends on the non-linearity of the two other contributions.

The limited impact of the non-linearity in the mean values, highlighted by the small values in Figure 4, shows that the responses to perturbation runs are robust. Indeed, this shows the sum of all contributions is equivalent to total $PM_{10}$ concentration. It is also important to note the non-linearity is slightly reduced by using the larger domains defining the cities (e.g. 9 grid cells), in a good agreement with the conclusions given by the country SC calculations (Pommier et al., 2020) and shown in Figure 3.

Figure 5 shows that this limited non-linearity impacts almost homogeneously all the cities in the "City" contributions, as noted with the color scale, with small exception over Malta, Tallinn, Reykjavik and in Switzerland. The Central European cities (e.g.

Berlin, Prague) are slightly more impacted by the non-linearity in the "Rest of Europe" and the "Extra sources" contributions. This is predictable due to the influence of the surrounding countries on their $PM_{10}$ over the relatively large area defining the cities (at least $0.25°$ longitude $\times$ $0.125°$ latitude). The non-linearity also varies from date to date over the cities (not shown).

This non-linearity remains limited, since in maximum, 7% of all the calculated hourly external contributions (Rest of Europe or Extra sources) for all 4-day forecasts over the selected cities have a non-linearity higher than 5% (0.1% for the City contribution – not shown).

## 5. Importance of the city contribution

### 5.1. Overview during the episode

Figure 6 shows the mean contribution of the "City" $PM_{10}$ on the total concentration for each city during this episode. To do so, we have calculated the mean ratio between the "City" concentration and the total $PM_{10}$ concentration for each date individually. Following the conclusions from Section 4, only the results related to a 15% reduction in the emissions and the city edges defined by 9 grids have been shown.

The surface background $PM_{10}$ over the Central European cities were not mainly impacted by the "City" sources which is
explained by the impact of the surrounding countries in these cities. This is also a good illustration of the statement given in Section 4 saying that the main contribution during the episode was from the "Rest of Europe", and essentially composed of "Domestic" country sources. Figure 7 shows this large impact of the "Domestic country" in the "Rest of Europe" contribution in most of the cities, except on the Central European cities and in Benelux impacted by the surrounding countries. Note that cities such as Nicosia and Valetta were mainly influenced by the "Extra sources" contribution which was essentially related to
natural sources and BCs.

Even if the city contribution was not the main contributor, cities such as Oslo and Lisbon, which did not experience large $PM_{10}$ concentrations, had a mean city contribution close to 70% on December 02nd and 03rd and close to 65% on December 5th, respectively (Fig. 6). A catalogue summarizing the mean of these hourly contributions for each individual day has been provided in the supplement. The three contributions (City, Rest of Europe and Extra Sources) are presented as well as the
Domestic country contribution. The catalogue also provides the information on the mean part of City in the Domestic Country contribution, the mean part of the Domestic Country in the Rest of Europe contribution and the $PM_{10}$ daily mean concentration. For Paris, the largest peaks are predicted on December 01st and on 02nd (e.g Fig. 2). On December 1st, the "City" contribution represented in average 44% of the $PM_{10}$ (see catalogue). On December $2^{nd}$, this decreased to 28% but continued to represent half of the "Domestic country" contribution. It is possible that the fraction of "city" $PM_{10}$ is underestimated, as the other
contributions, by the model. Indeed, in Pommier al. (2020), it has been shown that the regional model underestimates the larger hourly observed concentrations (see Section 4.1). This is predictable since a regional model, with a such resolution defining a

city, mainly captures the urban background concentrations which is not necessarily represented by the measurements in urban stations.

### 5.2. Complementary information with the country source apportionment: comparison between two cities

As illustration of the episode, a focus on the two large European cities has been decided, Paris and London. The comparison between both cities in their $PM_{10}$ concentrations highlights the possibility to use this source contribution calculations to understand the origin of the pollution. It may also help policy makers to identify a specific component which explains the concentration in $PM_{10}$ for a particular day. Figure 8 shows the main country contributors and the "City" contribution from 01 to 09 December 2016 predicted by the EMEP model over Paris while Figure 9 shows the results for London. The list of the

country contributors is related to the work done in Pommier et al. (2020) and corresponds to the 28 EU members plus Iceland, Norway and Switzerland as mentioned in Section 3.1. The other countries in the regional domain but not used in the country SC are gathered in the "External" contribution with the BCs.

It is worth noting that Paris had larger PM (fine and coarse), and $SO_x$ emissions during this period than London as shown in Table S3. At the opposite, London was characterized by larger CO and $NH_3$ emissions.

Large peaks in $PM_{10}$ over Paris and London have been calculated for the December 01st and 02nd (Figs. 8 and 9). These high concentrations over Paris mainly come from France with a large part coming from the city of Paris as predicted by the EMEP model (Fig. 8) while for the two first days over London, the $PM_{10}$ mainly have a British origin, external to London (Fig. 9). This British contribution represented in average 76% and 93% of the Rest of Europe contribution (or 62% and 75% of the total $PM_{10}$), on 01 December and 02 December, respectively (see catalogue). It is also clear with these figures 8 and 9 that London

was more influenced by external sources and by natural sources than Paris during this period.

During the two first days over Paris, the "City" contribution is attributed to primary components (rest PPM and EC, by 46% and 30% on Dec 01st and by 37% and 25% on Dec 2nd, respectively) as calculated by the EMEP model (Figs. S3 and S5). A report from the Paris regional air observatory (see https://www.airparif.asso.fr/_pdf/publications/pollution-episode-paris-area_dec2016.pdf) concluded the large $PM_{10}$ concentrations were mainly related to local sources such as wood burning and

traffic. Thus, Paris is a good illustration of the overall statement presented in Section 4.2. (Fig. 3a), concluding that the "City" contribution during the episode over the studied cities was dominated by the primary components. The importance of the primary components for this case also shows if the local emissions were reduced over this area during the 02 December, the level of urban background $PM_{10}$ could have been below the daily 50 ug/m$^3$ as recommended by WHO. For London, the EMEP model predicted that the British $PM_{10}$ was mostly due to $SO_4^{2-}$ (26%), showing that London has a different behavior than the

overall statement presented by Fig. 3b, where the "Rest of Europe" contribution was mainly due to $NO_3^-$ (even if $SO_4^{2-}$ is also an important contributor to the "Rest of Europe"). The part of primary component on the British $PM_{10}$ is larger for the following days when the British contribution to $PM_{10}$ is low.

## 6. Conclusions

This paper has presented the city source contribution product calculated by the EMEP model in a forecast mode and developed within the Copernicus Atmosphere Monitoring Service (CAMS). This product aims at identifying the sources responsible of the urban background $PM_{10}$ concentrations and this work has focused on an event occurring from 01 to 09 December 2016 over Europe. While the companion paper (Pommier et al., 2020) presented an evaluation of the calculation for the country contributions over 34 European cities, this paper has described the complementary information given by the prediction of the "City" contribution to the $PM_{10}$ concentrations in the same cities.

During the studied episode, 20% of the predicted $PM_{10}$ had a "City" origin, essentially composed of primary components, and 60% was from the countries in the regional domain (defined as "Rest of Europe"), essentially composed of $NO_3^-$ (by 35% and the two other secondary inorganic aerosols represent together ~30% of this contribution). This country contribution was mainly related to the Domestic country (e.g. Spain for Barcelona) (Pommier et al., 2020). The rest of the $PM_{10}$ was mainly due to natural sources. It was also shown that the Central European cities were mainly impacted by the surrounding countries while the cities located a little apart from the rest of the other European countries (e.g. Oslo and Lisbon) had a larger "City" contribution.

The methodology used in the EMEP model to calculate the contributions, has been based on perturbated emissions, known as a scenario approach. Thus, the change in the reduced emissions has been tested by using three different percentages: 5%, 15% and 50%. The definition of the city contribution, i.e. originating from the city itself and thus, the choice of the domain defining the edges of each studied city was also investigated. It was shown that the 15% reduction and the use of large city areas (9 grids or GADM) presented better results. The use of both parameters helps to prevent a larger impact of non-linearity in the chemistry, which is related to an assumed linear response in the concentrations due to changes in emissions. This non-linearity impacts the $NO_3^-$, $NH_4^+$ and H₂O concentrations. It was shown this non-linearity has a modest impact on the city contribution and essentially impacts the "Rest of Europe" contribution. For this contribution, the larger non-linearity (>5% of the total $PM_{10}$) represents only 7% of all the predicted hourly contributions over the different cities. This non-linearity has a slightly larger influence over the Central European cities for this "Rest of Europe" contribution, explained by the large impact of the surrounding countries, and thus from the different sources, on the urban $PM_{10}$ in these cities. The non-linearity may cause negative concentrations, but the negative contributions represented only less than 0.8% of the total concentrations. Other sources of uncertainties, such as the meteorological fields used for these predictions have not been addressed in this work. It is worth noting a good agreement has been found with meteorological observations over most of the cities. The use of more recent emission inventories such as CAMS-REG has also not been studied in this work.

The aim of the system is to predict in near-real time the urban and external contributions to the surface background $PM_{10}$ concentrations over different European cities, and it was shown the example of Paris has been a good illustration of the usefulness of the forecasting tool. The system has been able to predict the significant contribution from France and Paris as

well as the large impact of the primary components, during a polluted event occurring on 01 and 02 December 2016. It also confirms for this event that by reducing the emission of the local sources could help to reach the level below the recommended daily threshold established by the WHO. However, the city contribution as well as the other contributions presented in this work over the studied cities, may be underestimated on hourly resolution as suggested in the companion paper (Pommier et al., 2020). In this companion paper, it was shown the regional model underestimates the largest hourly urban concentrations

which is predictable due to the relatively coarse resolution used to define a city. An inter-comparison with another technique to estimate the urban background concentrations, or with another model by applying the same scenario approach have not been addressed in this work but it might be subject to another study by performing a full year evaluation. Moreover, details on the sectoral contribution, which is not provided in this work, should be an important information to further describe this episode.

**Data availability**

The EMEP model is an open source model available on https://doi.org/10.5281/zenodo.3355041. The data processing scripts are available on https://doi.org/10.5281/zenodo.4191038.

**Author contribution**

MP performed the experiment, analyzed the data and wrote the manuscript.


**Competing interests**

The author declares that there is no conflict of interest.

**Acknowledgments**

This work is partly funded by the EU Copernicus project CAMS 71 to provide policy support. This work has also received

support from the Research Council of Norway (Programme for Supercomputing) through the EMEP project (NN2890K) for CPU and the Norstore project "European Monitoring and Evaluation Programme" (NS9005K) for storage. The EMEP project itself is supported by the Convention on the Long Range Transmission of Air Pollutants, under UN-ECE. The author thanks H. Fagerli and M. Schulz (Norwegian Meteorological Institute) for involving him in the CAMS 71 project and giving him the opportunity to present the results. The author also thanks A. Valdebenito and A. Mortier (Norwegian Meteorological Institute),

for the development of the EMEP forecasting system; and for the development and the design of the website (https://policy.atmosphere.copernicus.eu/SourceContribution.php), respectively.

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

a)

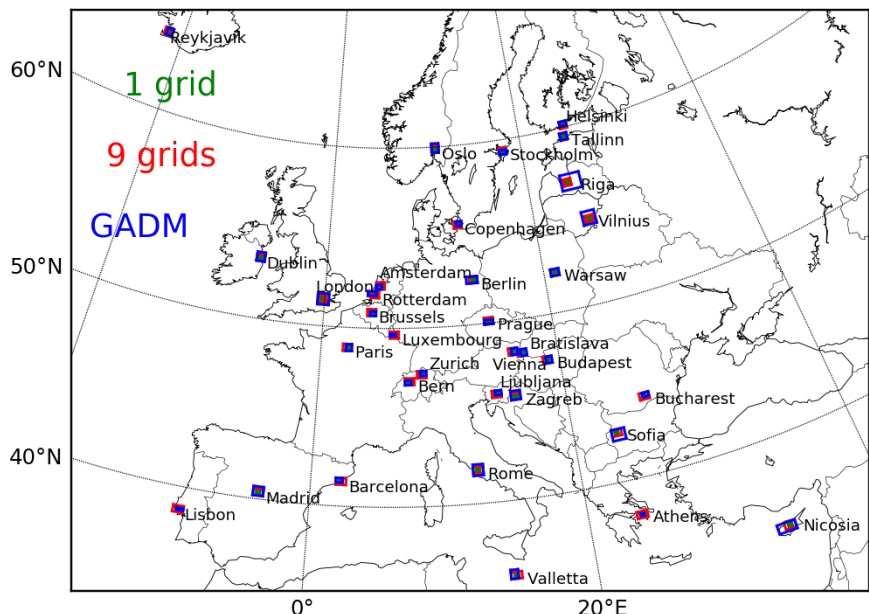

b)

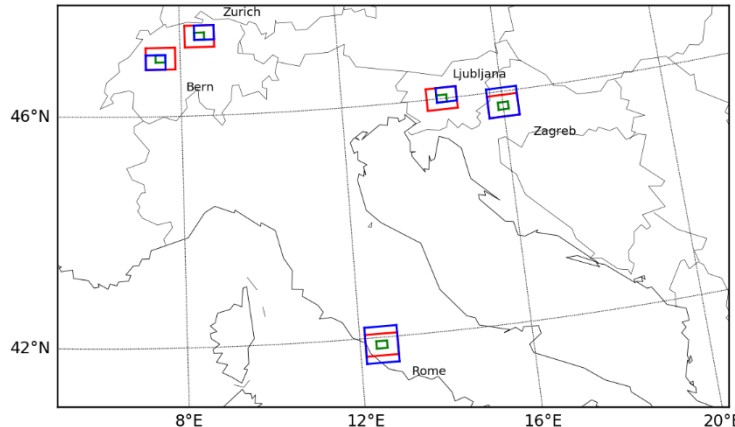

**Figure 1: a: Boxes defining each city edge, based on the 1 grid (green), 9 grids (red) and the GADM (blue) definitions, b: Zoom on a few cities highlighting the difference between the three definitions.**

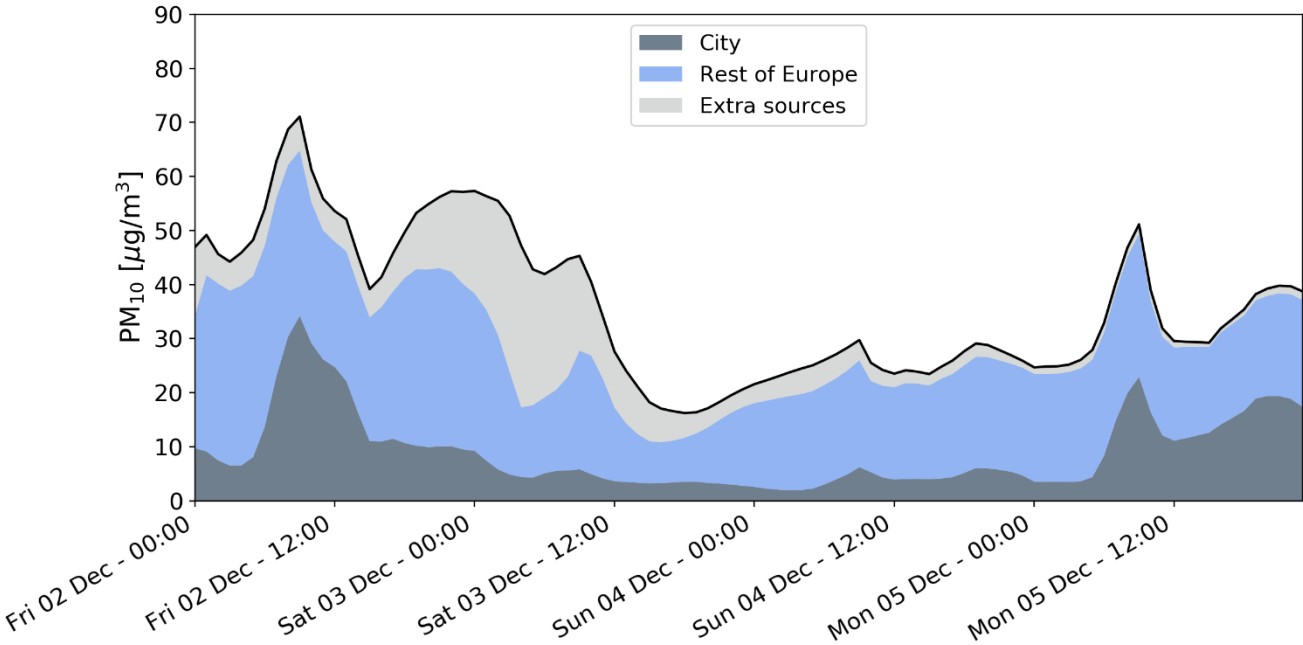

**Figure 2: Hourly PM$_{10}$ concentrations in μg/m$^3$ predicted by the EMEP model over Paris (defined by 9 grid cells) from 02 December to 05 December 2016. The black curve highlights the total concentration. The "city", "Rest of Europe" and "Extra sources" contributions are provided. "City" corresponds to the contribution from the area defined by 9 grid cells. "Rest of Europe" corresponds to all the European countries included in the regional domain and excluding the "City" contribution. "Extra sources" include the natural sources, the boundary conditions, the ship traffic, the biogenic sources, the soil NO emission, the aircraft emission and the lightning.**




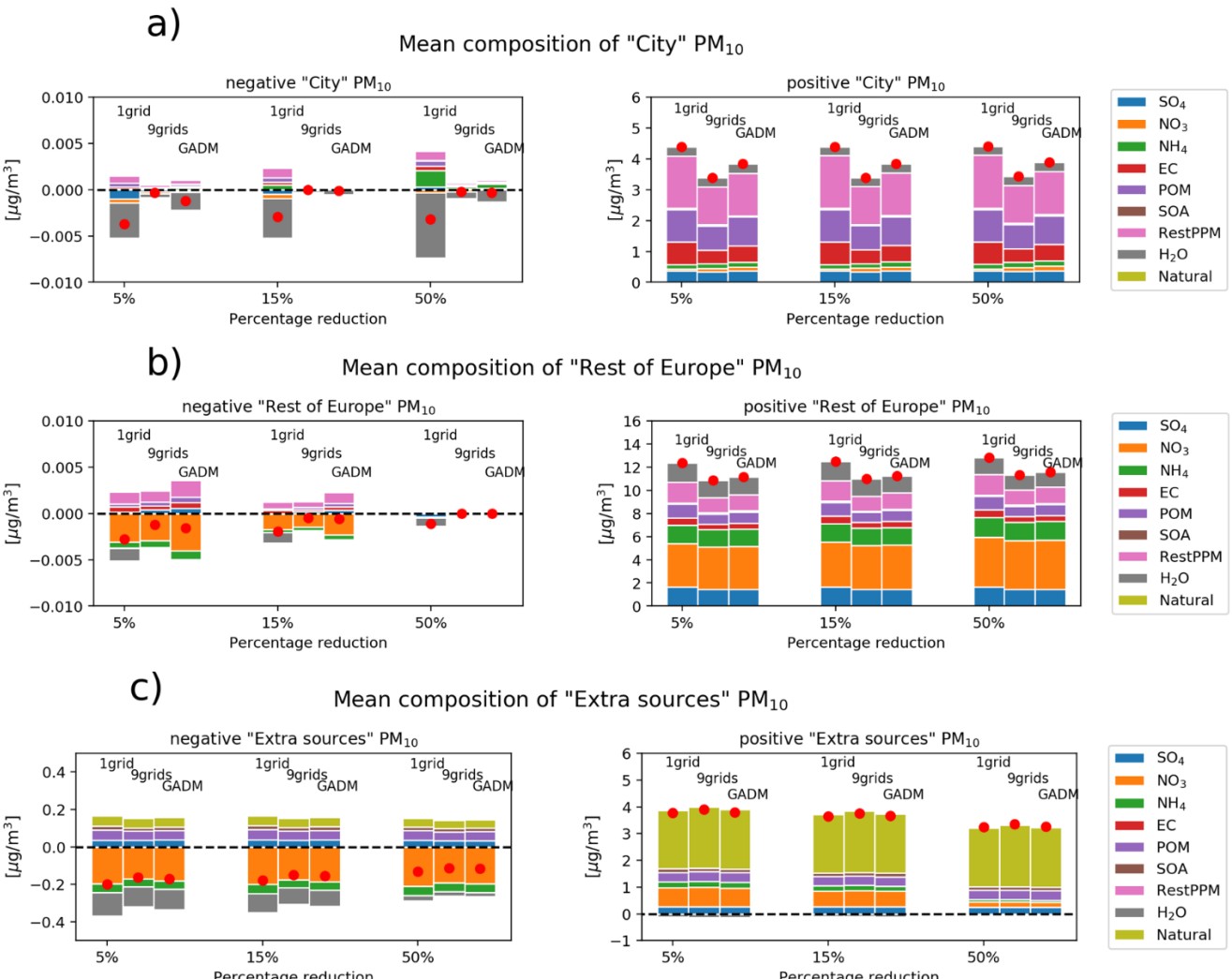

**Figure 3: Mean composition of "City" (a), "Rest of Europe" (b), and "Extra sources" PM₁₀ split into a negative concentration (left panel) and a positive concentration (right panel), calculated by the EMEP city source contribution over the 34 European cities and for each 4-day forecast. The PM₁₀ composition is highlighted with the color code. The results for the 3 city definitions (1 grid cell, 9 grid cells, GADM) and for the percentage of reduction used in the perturbation runs (5%, 15%, 50%) are shown. "Rest of Europe" corresponds to all the European countries included in the regional domain and excluding the "City" contribution. "Extra sources" include the natural sources, the boundary conditions, the ship traffic, the biogenic sources, the soil NO emission, the aircraft emission and the lightning. The red dot represents the mean PM₁₀ concentration.**

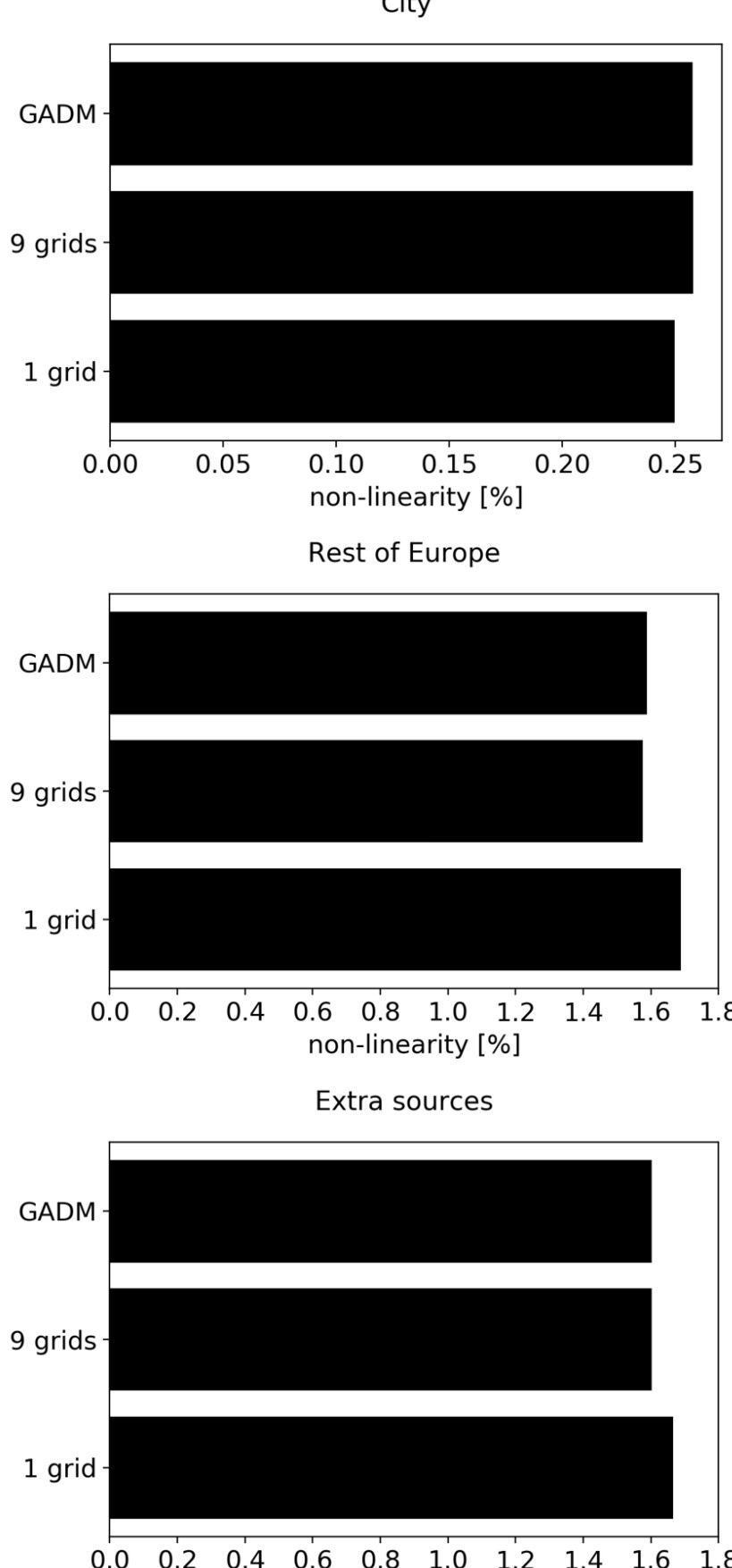

**Figure 4: The black horizontal bars show the mean non-linearity calculated for each contribution presented in Figure 3 and for the three city definitions. The non-linearity is calculated for each hourly concentration as the standard deviation of the hourly contribution weighted by the hourly mean of the total concentration.**


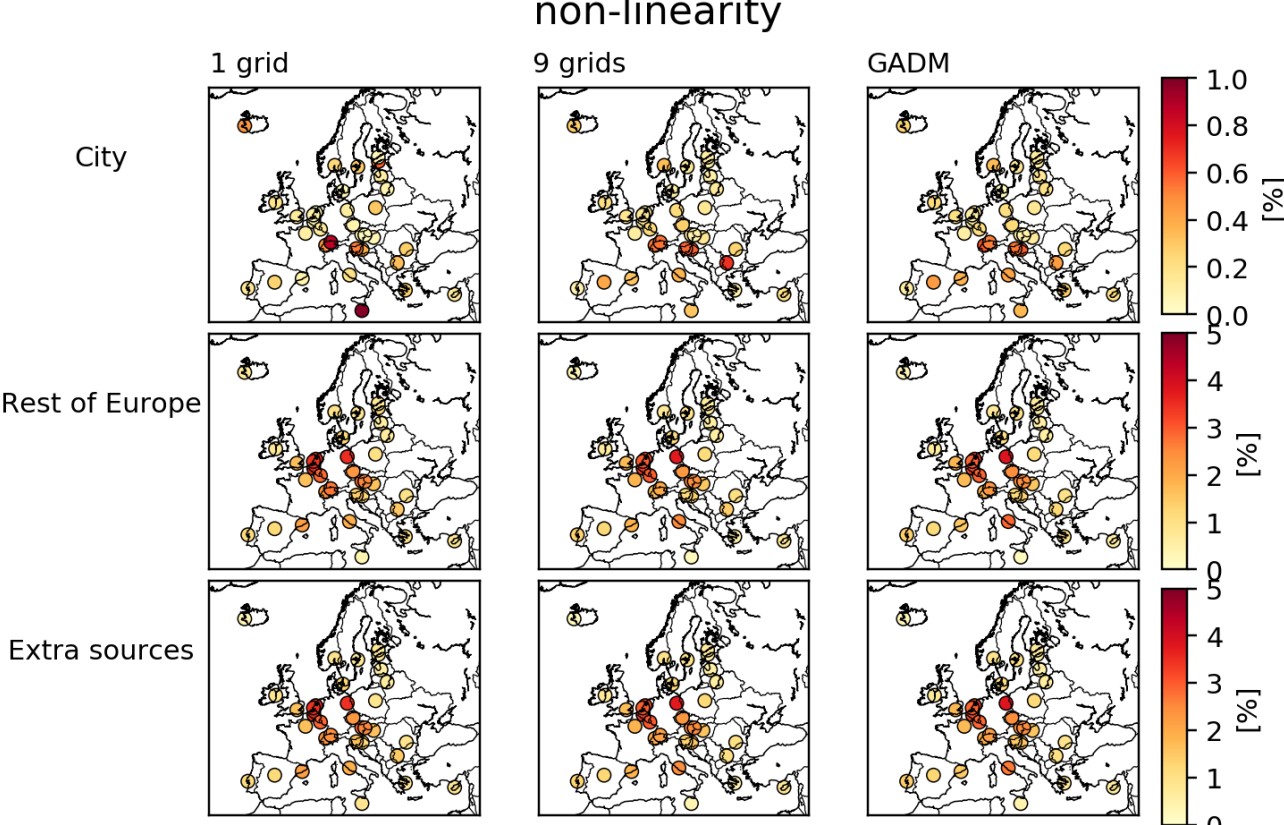

**Figure 5: Mean non-linearity in percent calculated for the "City", "Rest of Europe" and "Extra sources" contributions, over the 34 European cities and for each 4-day forecast (i.e. from 01-04 Dec to 09-12 Dec 2016). The non-linearity is presented for the cities defined by 1 grid (left row), 9 grids (middle row) and by the GADM (right row). Note the different scale to the "City" contribution compared to the two others.**



# 9 grids - pertubation factor: 15%

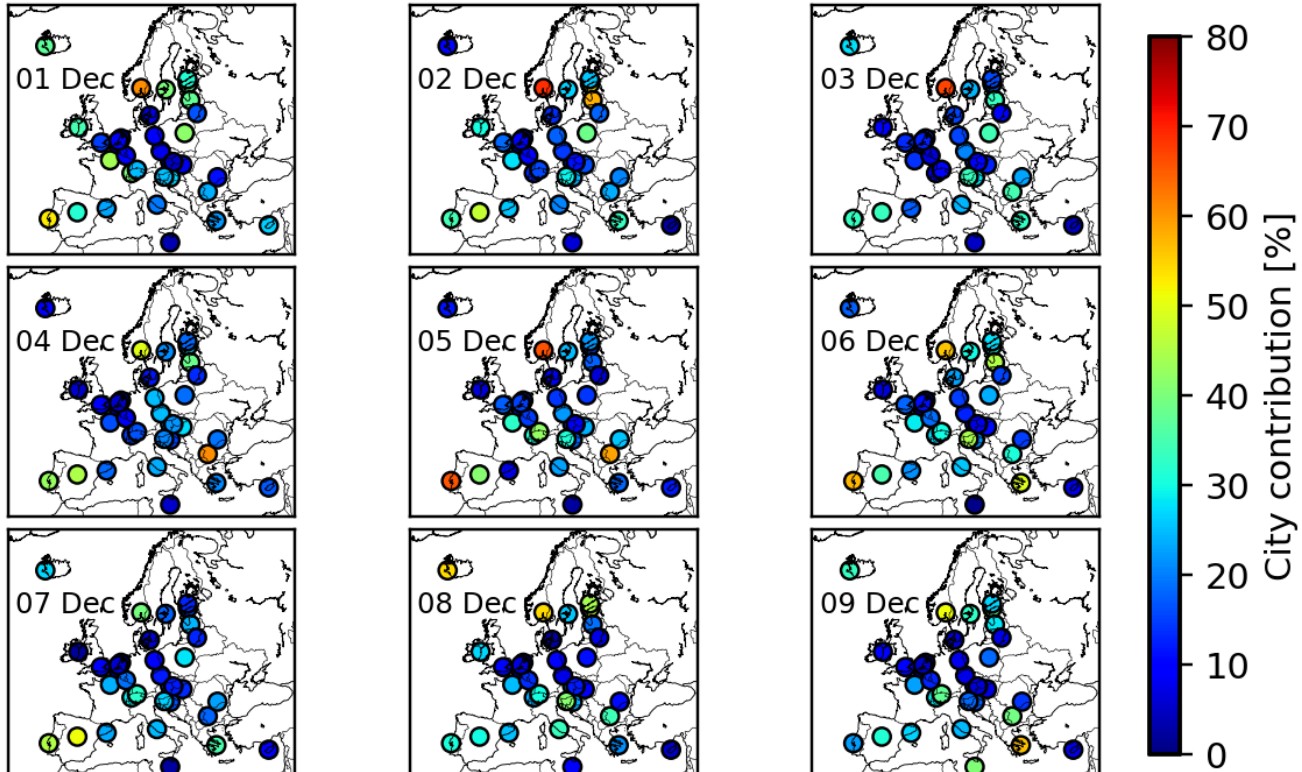

**Figure 6: Mean City contribution for each city from 01 to 09 December 2016. Each city edge is defined by 9 grid cells. The contribution is based on the calculations performed by the 15% perturbation runs.**


# 9 grids - pertubation factor: 15%

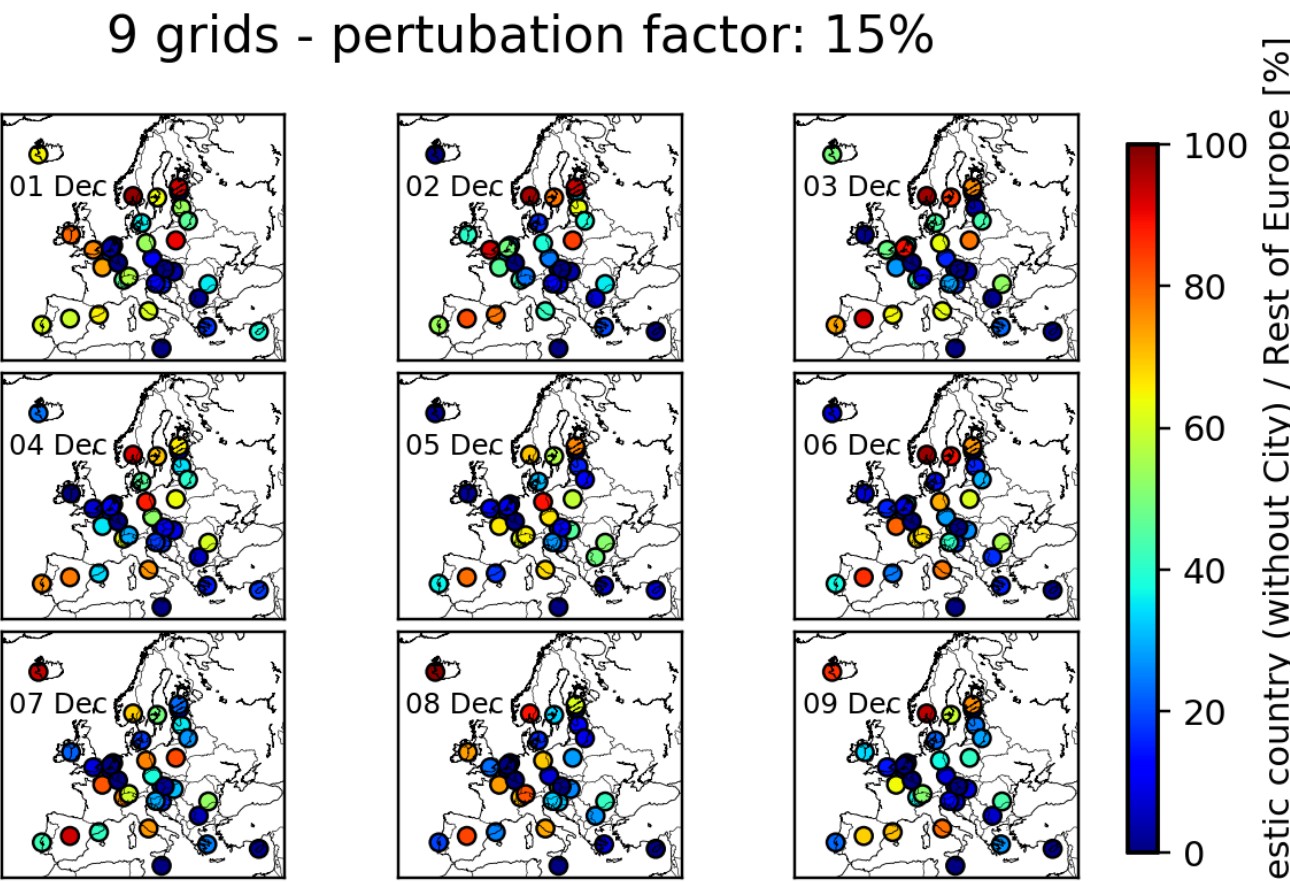

**Figure 7: Mean ratio of Domestic Country contribution (excluding the City contribution) to the Rest of Europe contribution in percent, for each city from 01 to 09 December 2016. Each city edge is defined by 9 grid cells. The contribution is based on the calculations performed by the 15% perturbation runs. The City contribution has been removed from Domestic Country contribution since it is not included into the Rest of Europe contribution.**



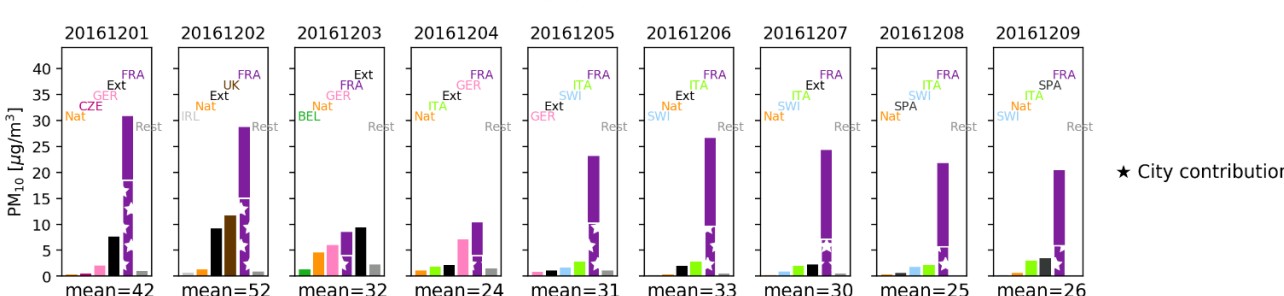

**Figure 8: Main country contributors to surface PM₁₀ over Paris, defined by 9 grid cells, for each single day from 01 to 09 December 2016 predicted by the EMEP model (see Pommier et al., 2020). The five main contributors are plotted. The "Rest" is the difference between the daily mean and the sum of these five contributors. The "external" contributor ("Ext" on the figure) essentially corresponds to the countries not included in the country SC runs and the BCs. The "City" contribution is highlighted by white stars. The daily mean surface PM₁₀ concentration is written below each bar chart. The labels BEL, CZE, FRA, GER, IRL, ITA, SPA, SWI, UK, refer to Belgium, Czechia, France, Germany, Ireland, Italy, Spain, Switzerland, and the United Kingdom, respectively.**

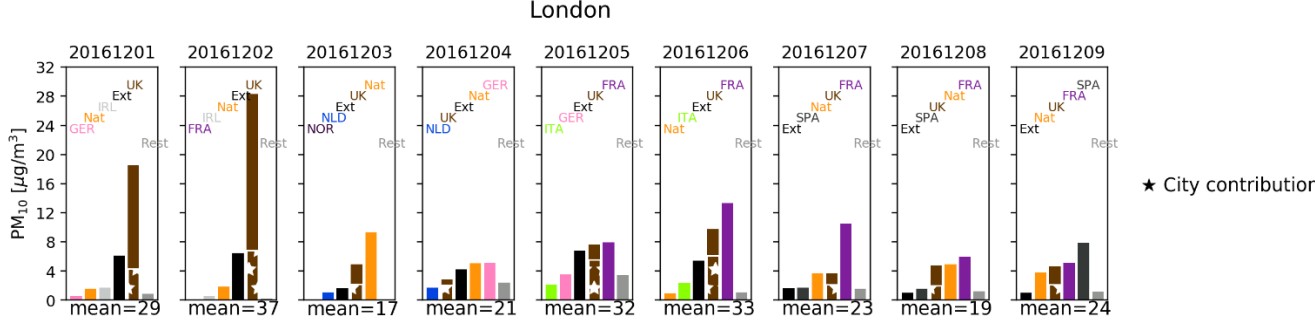

**Figure 9: As Fig. 8 for London. In addition to the previous labels, the labels NOR and NLD, correspond to Norway and the Netherlands, respectively**