# Peer review of "Prediction of source contributions to urban background PM10 concentrations in European cities: a case study for an episode in December 2016 using EMEP/MSC-W rv4.15 - Part.2 The city contribution"

_Geoscientific Model Development, 2020_

## Referee Comment (RC1) · Anonymous Referee #1 · 24 Nov 2020

General comments

In this paper, the author presents a methodology to predict the source contribution to surface PM10 concentrations for 34 European cities, focusing on the contribution from urban emissions. It is based on a set of EMEP model simulations run with a range of emission scenarios. This paper has a companion paper, Part 1 that is already published, focusing on the country contribution using a similar methodology and two

models. The present paper shows that the proposed methodology gives good results. In particular, it gives a detailed analysis of the possible effects of non-linearities on its performances. This methodology proves its robustness and is clearly of interest for discriminating the sources in case of regional PM10 events. The paper is well in the scope of the journal. The main weakness of the paper is the lack in some places of clear explanations or references to well support the methodology and associated finding.

Specific comments

Although this paper is the Part 2 of a series of papers and it is recommended to read Part 1 first, the understanding of Part 2 should be possible without. This is not the case because some important pieces of information from Part 1 are missing in Part 2 for a full understanding (see suggestions below).

In the title and in many other places in the paper, the adjective 'local' is used and sometimes together with background, which is somewhat contradictory. Since the EMEP model is run with a horizontal resolution that is coarse with respect to the urban scale, the model simulations cannot provide very local information (as discussed in the paper when compared to observations). I find the use of 'Local' is misleading, in particular in the abstract. I would recommend to use 'urban' rather than 'local' in the paper and remove local of the title.

Line 76: Here and in other places in the paper, there are references to the WHO exceedance limits which are limits related to human exposure. Because of the model resolution, the EMEP simulations do not represent the variations of the pollutant concentrations at the small scale within cities and therefore also not the actual exposure. Nevertheless, the methodology proposed on the basis of the EMEP model is useful to for characterizing regional air pollution events and the contribution of cities to these events. This needs to be clearly explained.

Lines 81-84. Secondary aerosols are generally not the main contributors to PM10

because they are mainly small particles. It would be useful that the author provides references on the relative contributions of secondary aerosols to PM10 in European cities, since secondary inorganic aerosols are largely discussed in the paper.

Line 84-85: 'these PM10 are essentially…... role'. This sentence needs to be supported by references. PM10 particles also undergo significant sedimentation due to their weight and dry deposition.

Introduction, 3rd paragraph where the Copernicus forecasting apportionment product and Part I paper are introduced. This paragraph is not clear: what means the different components? What the Part 1 paper is about and its link to the Copernicus product? How Part 2 is complementary to Part 1?

Introduction, 4th paragraph. An example is given of two methods assuming a linear relationship. Are there other methods not making this assumption?

Line 107: 'in cities in Europe" is too vague. Please, give a short description of the set of cities chosen.

Lines 112 to 120: The model resolution and its justification should be discussed first in this paragraph, before introducing the issue of the definition of city domain. About the resolution, I recommend to include a few sentences on the limitations related to the EMEP resolution for the present work but also for the previous work cited using similar model resolutions. Regarding the 3 definitions of the city domains, it is stated that they have been used in the companion paper but there is no information on what were the results from Part 1 on this particular point. More generally, a way to improve the introduction would be to give a short summary of the Part I objectives and results. This would be useful to understand what the present paper (Part 2) has in common and/or adds to Part 1.

Line 130: Could the author give a short description of the main updates?

Paragraph beginning line 131. The natural aerosols represented in EMEP need to be

included in the model description.

Lines 142-144: Same remark on wet deposition as in the introduction. Why is there a lot of details on wet deposition while there is no information on precipitation occurring during the pollution event?

Line 146-147: To understand the paragraph, change 'This estimate . . . precipitation' to This estimate is derived from large scale precipitation and convective precipitation accumulated at surface.'

Line 149-150: What is the value set for the precipitation intensity? On which basis this value is chosen?

Paragraph from line 168 to 174. The logical choice would be to use GADM which should represent the real extent of the cities. This is not clear to me why a 1 or 9 grid points should be tested. This needs to be argued. The '1 grid point' is obviously too small to represent most cities. The '9 grid points' assumes that the cities have an extension following a square shape which is not fully satisfactory, in particular for cities close to the seaside which then are assumed to extend over the sea.

Section 3.2: I am not sure to understand the first sentence. I think this is the chemistry (and/or possibly other processes) that causes non-linearities leading to errors in the method since the method assumes that the response to the change of anthropogenic emissions is linear. I also do not understand what the author means in the last two sentences. This section needs to be written more clearly.

Lines 225-226: What do the negative contributions correspond to? This is explained but later in the text. Figure 3 is a complex figure that requires detailed explanations.

Line 235: Is the 31st country not already one of the in the 30 European countries? Please clarify.

Line 242: Information on the results of the companion paper is needed here so that it makes easy to understand the combined results from Part 1 and Part 2.

[Figure]

Section 4.2 2nd paragraph: The author shows that $H_2O$ concentrations are impacted by the non-linearity. There is no explanation on what $H_2O$ refers to (relative humidity, concentration, vapour or liquid), except at the end of the paper where it seems that $H_2O$ is linked to relative humidity. In the EMEP simulation, the meteorological parameters come from IFS. In the troposphere, $H_2O$ mixing ratios are mainly driven by meteorology. Therefore, how $H_2O$ concentrations are affected by the change of anthropogenic emissions? Since changes associated to chemistry should be negligible with respect to the uncertainties in the meteorological water vapour field, where do the non-linearities come from? The treatment of $H_2O$ in EMEP model needs to be clearly explained in the model description in order to understand the analysis of the results.

Section 4.2 last paragraph: It would be useful to compare the uncertainties from the non-linearities to the model uncertainties that can be drawn from the comparison with the observations. This comparison has been done in Part 1 but this information has not been given in the present paper.

Section 5.2: The choice of the two cities (Paris and London) needs to be justified at the beginning of this section. Also, it would be useful to have the emissions for these cities for a more thorough interpretation of the results.

Lines 316-318: The author's analysis is consistent with an air quality report for Paris. Could the author explain what is the information in this report that supports his analysis?

Line 321: Reference to policies. The author may be more careful regarding policies since the present study assumes that the reduction of emissions applies to all sectors and with the same magnitude. In reality, policies on emissions during pollution events cannot be applied to all sectors and with the same level of regulation.

The conclusion section gives a summary of the methodology and results but little discussion on the other sources of uncertainties of the method than the non-linearity (for instance the meteorology or the parameterization of wet deposition) and also very few

prospects of extension of the methodology (for instance to PM2.5 or to the contribution of anthropogenic emissions by sector).

Technical corrections

Line 95: replace 'few' by 'a few'.

Line 148: Replace 'Precipitations are' by 'Precipitation is'.

Line 149: Replace 'precipitations occur' by 'precipitation occurs.

Line 165: Replace 'data set' by 'dataset'.

Figure 1 is small and it is very difficult to see the domains. More generally, most figures are small and uneasy to read.

Line 215: Replace 'was developed' by 'occurred'.

Line 299: Replace 'larger' by 'largest'.

Line 309: 'the list of country contributors are' to be replaced by 'the list of country contributors is'.

Line 310: Change 'correspond' to 'corresponds'.

Line 338: Change 'larger' by 'a larger'.

---

## Referee Comment (RC2) · Anonymous Referee #2 · 4 Dec 2020

This study estimated the source contribution of $PM_{10}$ concentrations based on a regional air quality forecasting model, and on a scenario approach in European cities. It was found that 20% of the predicted PM10 are from the city contributions (composed of primary PM components) and 60% are from the countries of the regional domain (excluding city contribution), and rest are contributed from the natural sources.

The major weakness of this study is the lack of the model evaluation in terms of the meteorological condition, PM10 concentrations, and the PM10 components. I think this information are strongly needed to ensure that the emission source contributions simulated from the model is reliable. In particular, this information is very important for designing the emission reduction plan in the future. Without these information, I don't think any of the model results can be trusted.

1. Can author state why the focus of the air pollution problem is PM10 instead of PM2.5? It seems nitrate contributes a lot for PM10 in your study region; however, I assumed most nitrate were the fine particles.

2. How serious is the PM10 problem in your study area? What is the characteristics of the PM10 in terms of seasonality and spatial variability? Is PM10 a serious air pollution problem in December?

3. There is lack of discussions of the emission distributions. A graphical demonstration of the emission distributions is very helpful for readers to have a good understanding of the PM10 problem in your study area.

4. There is a lack of the evaluation of the meteorological model performance and air quality model performance. Without this information, I don't think the model simulation results can be trusted.

5. There is a lack of the PM10 composition comparisons between the model and observation. This information is strongly needed to demonstrate that the model result is reliable and can be used to discuss the emission source contributions.

6. An overview of the study episode in terms of the observation characteristics

should be provided.

7. Line 165, the reference year of the anthropogenic emission data set is 2011. How representative is this old dataset when it is applied to discuss the current air quality conditions?

8. Section 4.1, there is a lack of the discussions of the observation characteristics. Fig. 2, there are only model simulation results which is not sufficient to persuade that the source contribution is reliable. A comparison with observation data is needed (e.g. Comparison with observed PM10 and PM10 components, to ensure the reliability of the model results).

9. Line 225, "the chemical reason of the non-linearity is revealed by the negative contributions to the predicted PM10 concentrations". Please clarify the sentence.

10. Line 230, "The mean PM10 concentration in a smaller area is larger, showing that with a smaller grid, the PM10 is less diffused over the integrated area." I think the discussions are weird. Isn't the 1-grid cell the closest grid to the emission source so that it has the largest concentrations?

11. Line 232, "The rest of Europe PM10 is mainly influenced by nitrate". Here, the nitrate concentration should be in the fine particle mode. Please provide evidence that demonstrating the PM compositions are mainly composed by nitrate.

    I am also curious why the nitrate occupy a large fraction of PM10 in Europe.

12. Line 249, why the nonlinearity only impact NO3, NH4 and H2O? What about SO2 and SO4?

13. Please explain why the eq 1 can be used to estimate the nonlinearity.

14. Line 315, what is the source of the EC and PPM?

15. Please provide evidence to support the SR result from model simulation.

16. It's not the reader's responsibility to read the PARTI of the companion study in order to understand this article. The author need to summarize the findings from the PARTI and explained in this study.

17. Use of "Local" contribution is very confusing.

18. Line 321, "if policies to reduce the local emissions over this area were performed during 02 December, the level of urban background PM10 would have been below the daily 50 ug/m$_3$". This statement needs to be supported with more evidence because it involves with the policy decision.

19. Section 5.2, the discussions in Paris and London should be evaluated with the observed data to support the findings.

20. Line 351-352, the source-contribution is done based on a scenario approach; however, due to various experiments are needed to be conducted which take amounts of computational time. How can this be accomplished in an air quality operational mode?

21. What is the main objective of this study? To provide the source contributions for PM10, or to develop a near-real time system that provides the source contributions to PM10? I don't think the design of the current study meet the study objective. For example, the scenario experiment does not provide comprehensive information of the source contributions. The discussions of developing the real-time source contribution technique are not introduced.

---

## Referee Comment (RC3) · Anonymous Referee #3 · 23 Dec 2020

For editor The author changed the definition of "Local" from country to city and produced another paper. There is nothing innovative in the current study but giving some valuable information that PM10 in most cities in Europe is mainly attributed to the area in domestic country in addition to the city. In addition, the author can provide information like "the contribution of "local", "domestic country in rest of Europe", "other countries in rest of Europe", and "Extra", in terms of each city. Therefore, the reviewer think it is worth to be read widely. However, there are some points the reviewer cares,
e.g., why the author used forecast meteorology instead of retrospective? the method of nonlinearity calculation, etc. Therefore, the reviewer suggests the outcome of this review is "major revision". The reviewer is willing to review the revised manuscript for the next submission.

For authors General comments: The reviewer used to think that the chemical non-linearity is the chemical reaction between sources. In this study, the authors used the ratio of standard deviation of hourly concentration to hourly concentration. What is the principle or base for their method? Why the nonlinearity is calculated based on statistics instead of chemistry? In addition, the authors cited Pommier et al. (2000) a lot. It is ok to cite a companion paper but the reader is not obligated to read the companion paper. Therefore, some information should be explained or mentioned in the current manuscript. At last, the reviewer thinks although the current study is not innovative compared with the Part I study but still provide a valuable information: "the contribution of "local", "domestic country in rest of Europe", "other countries in rest of Europe", and "Extra". Therefore, the reviewer suggests the author to list a table to provide such information in terms of every city. Special comments: 1. Please the model evaluation of meteorology, PM10, and PM10 compositions before any discussion in the manuscript. Readers are not obligated to read the Part I manuscript. Thus the authors should narrate or at least mention the model performance clearly. 2. On line 71, a comma or no blank between 400 and 000 is suggested. 3. On line 115, please explain the "concept" clearly. On line 116, please explain the meaning of "coarse". 4. Section 2.1, EMEP is not a meteorology-chemistry coupled model. Please supplement the description of meteorological inputs in current manuscript. 5. Section 2.2, is this study a forecast run or a retrospective run? Please narrate clearly. For such kind of study, a retrospective run is better than forecast run since the meteorology is the reanalysis data and closer to observations. 6. On line 166, please check the URL. The reviewer could not find the webpage. 7. On line 188, the choice of 15% is just because it is large enough to show clear concentration changes? Is there a stronger reason? Moreover, non-linearity represented less than 2% of total

concentrations for each predicted country contributions but may be larger for cities. Please reconsider your narratives. 8. The authors used zero-out emissions of two cities as a run. Is there any test that has proved hardly interaction exists between these two cities? 9. On line 193, please explain the method of perturbation run clearly. 10. On line 197, does the "Rest of Europe" include the domestic country? In other words, e.g., all areas in Europe in addition to Paris, right? 11. On line 198, "Then, this "Rest of Europe" contribution. . . . . . . .by the difference with the "Local" contribution" is suggested to "Then. . . . . . . .by the difference between the total and "Local contribution"". 12. On line 201, please narrate "scaled by 15%" more clearly. 13. On line 209, the reviewer could not understand the meaning of 9 "dates"? Is the simulation executed daily? Besides, "9 rest of EU", why?. What is the 9 reference runs? 14. On line 220, please give a strong reason why the reader is suggested to compare the Fig. 2 in current manuscript and Fig. 1 in in Pommier et al. (2020). 15. On line 223, please explain "4-d" predictions. 16. On line 239, what are the other sources (30-40%) for "Extra sources"? Are they the BCs? 17. Is the variance between city to city and date to date large? Is it proper to express in mean concentrations? 18. On line 241, please calculated the proportion of the "local", "Domestic country" in "Rest of Europe", "Rest of Europe" not including the "Domestic country", for example, Paris. 19. On line 246, the chemically non-linear effect is negative. Please denote the negative term is which minus which. On line 251, if NH4NO3 is formed by NOx and NH3 in different regions, there is additional PM10 formed. Therefore, the non-linear effect is positive, isn't it? 20. On line 256, "If this NOx is emitted in excess", why do the authors use "If" in this sentence? 21. On line 262, "it is very small", what is "it", "the impact of the percentage" or "the size of the city edges"? 22. On line 271, Please explain the formula (1) is reasonable and persuasive. 23. On line 271, n=3, is that representing standard deviation reliable in the view of statistics? 24. On line 275, "It is worth noting . . . . . .other contributions". Please explain clearly. 25. On line 277, "The limited impact of . . . . .are robust". Please explain clearly. 26. On line 298, the superscript is not needed for dates. 27. On line 300, the underestimated hourly PM2.5 doesn't

mean the "Local" PM2.5 is also underestimated. The underestimation could be due to other sources. 28. Fig. 7, Fig. 8, please denote the full name of countries. Not everyone understands the abbreviations of countries. 29. Fig. 7 captions, what is "countries not included in the country SC runs"? Please explain it clearly in the current manuscript. 30. Fig. 7 captions, "The five main contributors are plotted as well as the difference between the daily mean and the sum of these five contributors ("Rest").". This sentence should be split to two sentence: "The five main contributors are plotted as well." and "The "Rest is the difference between the daily mean and the sum of these five contributors", right?

Please also note the supplement to this comment:
https://gmd.copernicus.org/preprints/gmd-2020-242/gmd-2020-242-RC3-supplement.pdf

---

## Author Comment (AC1) · 27 Mar 2021

The comment was uploaded in the form of a supplement:
https://gmd.copernicus.org/preprints/gmd-2020-242/gmd-2020-242-AC1-
supplement.zip

---

## Author Comment (AC2) · 27 Mar 2021

The comment was uploaded in the form of a supplement: https://gmd.copernicus.org/preprints/gmd-2020-242/gmd-2020-242-AC2-supplement.zip

---

## Author Comment (AC3) · 27 Mar 2021

The comment was uploaded in the form of a supplement:
https://gmd.copernicus.org/preprints/gmd-2020-242/gmd-2020-242-AC3-supplement.zip

---

## Author Response (AR1)

**Reviewer 1:**

**General comments**

In this paper, the author presents a methodology to predict the source contribution to surface PM10 concentrations for 34 European cities, focusing on the contribution from urban emissions. It is based on a set of EMEP model simulations run with a range of emission scenarios. This paper has a companion paper, Part 1 that is already published, focusing on the country contribution using a similar methodology and two models. The present paper shows that the proposed methodology gives good results. In particular, it gives a detailed analysis of the possible effects of non-linearities on its performances. This methodology proves its robustness and is clearly of interest for discriminating the sources in case of regional PM10 events. The paper is well in the scope of the journal. The main weakness of the paper is the lack in some places of clear explanations or references to well support the methodology and associated finding.

**Specific comments**

Although this paper is the Part 2 of a series of papers and it is recommended to read Part1 first, the understanding of Part2 should be possible without. This is not the case because some important pieces of information from Part 1 are missing in Part 2 for a full understanding (see suggestions below).

I would like to start to thank the reviewer 1 for his careful reading and his comments. I apologize for the missing information I did not provide in the previous version of the paper. I hope I have solved this issue and I agree I should have provided most of the information in the previous version. I am sure the corrections help to improve this study. I have answered all the points by writing my reply in blue.

Among the corrections, the revised manuscript provides:
- more information on the evaluation of the predictions (on $PM_{10}$ and meteorological fields)
- more details on the episode
- clarification on the chemical non-linearity
- information on the emissions
- a catalogue of the different contributions for each city and for each day.

In the title and in many other places in the paper, the adjective 'local' isused and sometimes together with background, which is somewhat contradictory. Since the EMEP model is run with a horizontal resolution that is coarse with respect to the urban scale, the model simulations cannot provide very local information (as discussed in the paper when compared to observations). I find the use of 'Local' is misleading, in particular in the abstract. I would recommend to use 'urban' rather than 'local' in the paper and remove local of the title.

The idea was to split the background concentration over urban area from local sources and external sources.

However, I agree the term of "local urban background" can be confusing. Thus, the word "local" has been changed by "city". Moreover, the title has also been changed to be consistent with the Part 1.

Now it reads:

"Prediction of source contributions to urban background $PM_{10}$ concentrations in European cities: a case study for an episode in December 2016 using EMEP/MSC-W rv4.15 - Part.2 The city contribution".

Line 76: Here and in other places in the paper, there are references to the WHO exceedance limits which are limits related to human exposure. Because of the model resolution, the EMEP simulations do not represent the variations of the pollutant concentrations at the small scale within cities and therefore also not the actual exposure. Nevertheless, the methodology proposed on the basis of the EMEP model is useful to for characterizing regional air pollution events and the contribution of cities to these events. This needs to be clearly explained.

It is correct. I have modified a sentence at the end of the introduction, highlighting this study focuses on background concentration.

The sentence is:

"It is worth noting, the definition uses a relatively coarse resolution (at least 0.25° longitude × 0.125° latitude) which is representative of the background concentration, and is comparable to the definition of the city domain used in previous studies such as in Thunis et al. (2016) who used an area of $35 \times 35$ km$^2$ or in Skyllakou et al. (2014) who used a radius of 50 km from the city center."

Lines 81-84. Secondary aerosols are generally not the main contributors to PM10 because they are mainly small particles. It would be useful that the author provides references on the relative contributions of secondary aerosols to PM10 in European cities, since secondary inorganic aerosols are largely discussed in the paper.

I agree, and the following sentence has been added:

"These secondary aerosols can represent a large fraction of the PM$_{10}$ composition in European cities (e.g. Querol et al., 2004; Amato et al., 2016; Redington et al., 2016, Diapouli et al., 2017)."

With the corresponding references:

- Amato, F., Alastuey, A., Karanasiou, A., Lucarelli, F., Nava, S., Calzolai, G., Severi, M., Becagli, S., Gianelle, V. L., Colombi, C., Alves, C., Custódio, D., Nunes, T., Cerqueira, M., Pio, C., Eleftheriadis, K., Diapouli, E., Reche, C., Minguillón, M. C., Manousakas, M.-I., Maggos, T., Vratolis, S., Harrison, R. M., and Querol, X.: AIRUSE-LIFE+: a harmonized PM speciation and source apportionment in five southern European cities, Atmos. Chem. Phys., 16, 3289–3309, https://doi.org/10.5194/acp-16-3289-2016, 2016.
- Querol, X., Alastuey, A., Ruiz, C.R., Artíñano, B., Hansson, H.C., Harrison, R.M., Buringh, E., Brink, H.M. ten, Lutz, M., Bruckmann, P., Straehl, P., Schneider, J.: Speciation and origin of PM10 and PM2.5 in selected European cities. Atmos. Environ. 38, 6547–6555, https://doi.org/10.1016/j.atmosenv.2004.08.037, 2004.
- Redington, A. L., Witham, C. S., Hort, M. C.: Source apportionment of speciated PM$_{10}$ in the United Kingdom in 2008: Episodes and annual averages, Atmos. Env., 145, 251-263, https://doi.org/10.1016/j.atmosenv.2016.09.047, 2016.
- Diapouli, E., Manousakas, M., Vratolis, S., Vasilatou, V., Maggos, Th, Saraga, D., Grigoratos, Th, Argyropoulos, G., Voutsa, D., Samara, C., Eleftheriadis, K.: Evolution of air pollution source contributions over one decade, derived by PM$_{10}$ and PM$_{2.5}$ source apportionment in two metropolitan urban areas in Greece, Atmos. Env., 164, 416-430, https://doi.org/10.1016/j.atmosenv.2017.06.016, 2017.

Line 84-85: 'these PM10 are essentially..... role'. This sentence needs to be supported by references. PM10 particles also undergo significant sedimentation due to their weight and dry deposition.

The following references have been added:

Fuzzi, S., Baltensperger, U., Carslaw, K., Decesari, S., Denier van der Gon, H., Facchini, M. C., Fowler, D., Koren, I., Langford, B., Lohmann, U., Nemitz, E., Pandis, S., Riipinen, I.,

Rudich, Y., Schaap, M., Slowik, J. G., Spracklen, D. V., Vignati, E., Wild, M., Williams, M., and Gilardoni, S.: Particulate matter, air quality and climate: lessons learned and future needs, Atmos. Chem. Phys., 15, 8217–8299, https://doi.org/10.5194/acp-15-8217-2015, 2015.

And

Mitchell, R., Maher, B.A., Kinnersley, R.: Rates of particulate pollution deposition onto leaf surfaces: Temporal and inter-species magnetic analyses, Env. Poll., 158, 5, 1472-1478, https://doi.org/10.1016/j.envpol.2009.12.029, 2010.

Introduction, 3rd paragraph where the Copernicus forecasting apportionment product and Part I paper are introduced. This paragraph is not clear: what means the different components? What the Part 1 paper is about and its link to the Copernicus product? How Part 2 is complementary to Part 1?

The paragraph is now:

"To provide information to identify the sources of the polluted events over different European cities, a forecasting source apportionment product has been developed within the Copernicus Atmosphere Monitoring Service (CAMS). The predictions are calculated for 4 days and are available on the website https://policy.atmosphere.copernicus.eu/SourceContribution.php. The calculations are provided for the surface $PM_{10}$ and its different components over European cities. The predictions are done as a complement to the country source contribution calculations, providing information on the countries responsible of the same polluted events. These country contributions are described in a companion paper (Pommier et al., 2020). The calculations, presented in this study, separate the city contribution from external contributions. Thus, by combining the information from the country contribution given in the companion paper and the city contribution presented hereafter, the system allows providing information on long-range transport in the European cities and the pollution coming from the urban area. These contributions might be important to determine short term air pollution control measures, which can remain difficult to assess by local authorities."

Introduction, 4th paragraph. An example is given of two methods assuming a linear relationship. Are there other methods not making this assumption?

In the scenario approach, yes, the linear relationship is assumed.

However, other methodologies, such as the labelling technique used in LOTOS-EUROS and the similar approach of the tagging system used in CMAQ do not assume a linear relationship between the reduced emissions and the changes in concentrations.

The labelling technique tracks inert aerosol tracers as well as chemically active tracers containing a C, N (reduced and oxidised) or S atom through chemical reactions within an unchanged chemical regime (Kranenburg et al., 2013).

The tagging system uses additional species which provides information on contributions on group of sources, sectors etc. but does not react to changes in emissions. At each time step in the simulation, the effects of processes, such as dry deposition and advection, are calculated directly for all tagged species (Douglas et al. 2006).

Kranenburg, R., Segers, A. J., Hendriks, C., and Schaap, M.: Source apportionment using LOTOS-EUROS: module description and evaluation, Geosci. Model Dev., 6, 721-733, https://doi.org/10.5194/gmd-6-721-2013, 2013.

Douglas, S., T. Myers and Y. Wei. 2006. " Implementation of Sulfur and Nitrogen Tagging in the Community Multiscale Air Quality (CMAQ) Model." Prepared for EPA, OAQPS, Research Triangle Park, NC. ICF International, San Rafael, California (06-076).

Line 107: 'in cities in Europe" is too vague. Please, give a short description of the set of cities chosen.
The mentioned issue is relevant for all European cities so there is no specification in this sentence. However, I agree the information on the selected cities studied in this work was missing. The following has been modified as below:
"Thus, the objective of this study is to present the calculation of the **near-real time urban background** contribution predicted by the EMEP/MSC-W model on hourly resolution **for each capital of the 28 European Union countries plus Barcelona, Bern, Oslo, Reykjavik, Rotterdam and Zurich**."

Lines 112 to 120: The model resolution and its justification should be discussed first in this paragraph, before introducing the issue of the definition of city domain. About the resolution, I recommend to include a few sentences on the limitations related to the EMEP resolution for the present work but also for the previous work cited using similar model resolutions. Regarding the 3 definitions of the city domains, it is stated that they have been used in the companion paper but there is no information on what were the results from Part 1 on this particular point. More generally, a way to improve the introduction would be to give a short summary of the Part I objectives and results. This would be useful to understand what the present paper (Part 2) has in common and/or adds to Part 1.

To clarify the objective, the following sentences have been added:
- "Thus, by combining the information from the country contribution given in the companion paper and the city contribution presented hereafter, the system allows providing information on long-range transport in the European cities and the pollution coming from the urban area. These contributions might be important to determine short term air pollution control measures, which can remain difficult to assess by local authorities."
and
- "Pommier et al. (2020) have already shown this event was mainly related to emissions of the Domestic country, i.e. coming from the country corresponding to the studied city such as France for Paris, while the influence of other countries was mainly characterized by a large fraction of $NO_3^-$. However, the contribution from the city, included in this Domestic Country contribution, was not estimated in this companion paper."

Moreover, the mentioned paragraph has been changed:
"For the calculation of this "City" contribution, the definition of the city area is a critical parameter. For this reason, the domain defining the studied cities was investigated. It is worth noting, the definition uses a relatively coarse resolution (at least 0.25° longitude × 0.125° latitude) which is representative of the background concentration, and is comparable to the definition of the city domain used in previous studies such as in Thunis et al. (2016) who used an area of 35 × 35 km² or in Skyllakou et al. (2014) who used a radius of 50 km from the city center. Thus, 1 model grid cell (0.25° longitude × 0.125° latitude), 9 grid cells and the grid cells covering the definition given by the Global Administrative Area - GADM) have been used as also done in Pommier al. (2020). Pommier et al. (2020) found by using a larger domain defining the cities helps to limit the impact of the chemical non-linearity in the predictions. In this work, the "City" contribution corresponds to the averaged concentration over a studied

city. It is worth noting in our definition of the "city" contribution, there is no distinction between the urban background and the rural background which both may impact the concentration of the pollutant over a city as explained in Thunis et al. (2018)."

Line 130: Could the author give a short description of the main updates?
The following sentences with the corresponding references have been added:
"The main updates since the version presented in Simpson et al. (2012) and used in this work, concern a new calculation of aerosol surface area (now based upon the semi-empirical scheme of Gerber, 1985), a revised parameterizations of $N_2O_5$ hydrolysis on aerosols, an additional gas-aerosol loss processes for $O_3$, nitric acid ($HNO_3$) and hydroperoxy radical ($HO_2$), a new scheme for ship $NO_x$ emissions, a new calculated natural marine emissions of dimethyl sulphid (DMS), the use of a new land-cover (used to calculate biogenic VOC emissions and the dry deposition) and an update in the source function for sea salt production to account for whitecap area fractions, following the work of Callaghan et al. (2008) (Simpson et al., 2016 and 2017)."

- Callaghan, A., de Leeuw, G., Cohen, L., and O'Dowd, C. D.: Relationship of oceanic whitecap coverage to wind speed and wind history, Geophys. Res. Lett., 35, L23609, https://doi.org/10.1029/2008GL036165, 2008.
- Gerber, H. E.: Relative-Humidity Parameterization of the Navy Aerosol Model (NAM), Naval Research Laboratory, NRL report 8956, 1985.
- Simpson, D., Nyíri, Á, Tsyro, S., Valdebenito, Á and Peter Wind and Wind, P.: Updates to the EMEP/MSC-W model, 2015-2016 Transboundary particulate matter, photo-oxidants, acidifying and eutrophying components. EMEP Status Report 1/2016, The Norwegian Meteorological Institute, Oslo, Norway, 15-36, ISSN 1504-6109, 2016.

Paragraph beginning line 131. The natural aerosols represented in EMEP need to be included in the model description.
In addition to the information provided in the updates (see the reply to the previous comment), the following sentence has been added:
"The sea salt generation is based on two source functions, those of Monahan et al. (1986) and Mårtensson et al. (2003) as described in Tsyro et al. (2011)."

- Mårtensson, E.M., Nilsson, E.D., de Leeuw, G., Cohen, L.H. and Hansson, H.C.: Laboratory simulations and parameterization of the primary marine aerosol production. J. Geophys. Res. Atmos., 108., D9, 4297, doi:10.1029/2002JD002263, 2003.
- Monahan, E., Spiel, D., and Davidson, K.: A model of marine aerosol generation via white caps and wave disruption, in: Oceanic whitecaps, edited by: Monahan, E. and MacNiochaill, G., 167–193, Dordrecht: Reidel, The Netherlands, 1986.
- Tsyro, S., Aas, W., Soares, J., Sofiev, M., Berge, H., and Spindler, G.: Modelling of sea salt concentrations over Europe: key uncertainties and comparison with observations, Atmos. Chem. Phys., 11, 10367–10388, https://doi.org/10.5194/acp-11-10367-2011, 2011.

Lines 142-144: Same remark on wet deposition as in the introduction. Why is there a lot of details on wet deposition while there is no information on precipitation occurring during the pollution event?
There are more details about the meteorological conditions in section on the description of the episode:

"In December 2016, a PM episode occurred across North-Western Europe, as a consequence of a high-pressure system Europe (see http://policy.atmosphere.copernicus.eu/reports/CAMSReportDec2016-episode.pdf). December 2016 was one of the warmer Decembers that Europe has ever known. For example, The United Kingdom reported its eighth warmest December in a series dating to 1910. In Norway, December temperature was 4.6°C above its 1961–1990 national average, making this one of the 10 warmest Decembers in the country's 117-year period of record. In a same time, December 2016 was drier than the normal, except in Norway. France was record dry, with average precipitation totals only 20 percent of its 1991–2010 average, breaking the previous record low of December 2015, and Austria had the driest December, where precipitation records date back to 1851 (NOAA, Global Climate Report for December 2016).

Line 146-147: To understand the paragraph, change 'This estimate ... precipitation' to This estimate is derived from large scale precipitation and convective precipitation accumulated at surface.'
It has been changed as requested.

Line 149-150: What is the value set for the precipitation intensity? On which basis this value is chosen?
This sentence means the precipitation is homogeneously distributed on the vertical.
It is done in the met_ml.f90 routine of the EMEP model.

Paragraph from line 168 to 174. The logical choice would be to use GADM which should represent the real extent of the cities. This is not clear to me why a 1 or 9 grid points should be tested. This needs to be argued. The '1 grid point' is obviously too small to represent most cities. The '9 grid points' assumes that the cities have an extension following a square shape which is not fully satisfactory, in particular for cities close to the seaside which then are assumed to extend over the sea.
It is correct and it was a long discussion between partners of project during the development of the forecasting tool.
Even if the use of the grid cells covering the GADM definition corresponds to the area which is most accurate, there is still the issue of the square shape. In our GADM city definition, it uses all the grid cells overlapping the GADM boundaries.
Moreover, to use a common area for all cities help to have an unified comparison between all cities. However, I agree that the three definitions which are tested have their pros and cons.

Section 3.2: I am not sure to understand the first sentence. I think this is the chemistry (and/or possibly other processes) that causes non-linearities leading to errors in the method since the method assumes that the response to the change of anthropogenic emissions is linear. I also do not understand what the author means in the last two sentences. This section needs to be written more clearly.
This part has been rephrased as below:
"As explained previously, the calculated concentrations based on a scenario approach, may be impacted by non-linearity. The calculated concentrations due to a reduced emission depend on the atmospheric composition already presents. The total $PM_{10}$ over the receptor should be theoretically identical to the sum of the $PM_{10}$ originated from the different sources, but due to this non-linearity, this is not always the case and it might have few differences between the total $PM_{10}$ and the sum from the various sources."

Lines 225-226: What do the negative contributions correspond to? This is explained but later in the text. Figure 3 is a complex figure that requires detailed explanations.

These following sentences have changed as below:

"The sum of each contribution should correspond to the total $PM_{10}$ calculated by the reference run, but some differences can appear. By splitting the $PM_{10}$ concentrations for each contribution based on their sign, the negative $PM_{10}$ concentrations help to reveal the species impacted by the non-linearity and explaining the differences seen with the total $PM_{10}$ concentrations calculated by the reference run. On the other hand, the positive concentrations provide the information on the overall composition for each contribution."

Line 235: Is the 31st country not already one of the in the 30 European countries? Please clarify.

I agree it was not clear. I have changed the sentence and the information is now provided in Section 3.1:

"Thus, all these simulations are a complementary information of the country contributions presented in Pommier al. (2020). Indeed, in the country contribution calculations provided in Pommier et al. (2020), there is the "Domestic country" which represents the country corresponding to the studied city (e.g. Spain for Barcelona). Another contributor in the country SC is "30 European countries". In the country SC, the contributions for 31 countries are calculated which include the 28 EU countries, Iceland, Norway and Switzerland, and this "30 European countries" combines all these contributors and excludes the "Domestic country"."

Line 242: Information on the results of the companion paper is needed here so that it makes easy to understand the combined results from Part 1 and Part 2.

Please note, the following sentence has been added in the Introduction:

"Pommier et al. (2020) have already shown this event was mainly related to emissions of the Domestic country, i.e. coming from the country corresponding to the studied city such as France for Paris, while the influence of other countries was mainly characterized by a large fraction of $NO_3^-$. However, the contribution from the city, included in this Domestic Country contribution, was not estimated in this companion paper."

Moreover, the section 4.1 has been rewritten with additional figures in a supplement. Among these corrections, there is this new paragraph:

"It shows that the main contributor of the $PM_{10}$ during the episode was caused by the long-range transport. Since there is a low contribution from cities, and the country SC showed that the main contributor was the "domestic country", that means the "Rest of Europe" contribution is mainly composed of this "domestic country". In other words, that means this episode was mainly influenced by the "Domestic" country and not by the cities."

Section 4.2 2nd paragraph: The author shows that H2O concentrations are impacted by the non-linearity. There is no explanation on what H2O refers to (relative humidity, concentration, vapour or liquid),except at the end of the paper where it seems that H2O is linked to relative humidity. In the EMEP simulation, the meteorological parameters come from IFS. In the troposphere, H2O mixing ratios are mainly driven by meteorology. Therefore, how H2O concentrations are affected by the change of anthropogenic emissions? Since changes associated to chemistry should be negligible with respect to the uncertainties in the meteorological water vapour field, where do the non-linearities come from? The treatment of H2O in EMEP model needs to be clearly explained in the model description in order to understand the analysis of the results.

H2O corresponds to the concentration.

To be clearer, additional information has been added in Section 2.1, on the description of the model:

"The EMEP model uses the MARS equilibrium module of Binkowski and Shankar (1995) to calculate the partitioning between gas and fine-mode aerosol phase in the system of $SO_4^{2-}$-$HNO_3$-$NO_3^-$-$NH_3$-$NH_4^+$. This module also calculates the mass of aerosol water (Simpson et al., 2012). This calculated mass of water is added to dry PM$_{10}$ masses when being compared with measured concentrations."

This explains why H$_2$O is impacted by non-linearity since it is involved in the gas portioning of SIA.

In the 1$^{st}$ paragraph in Section 4.2, the information in bold has also been added:
"This impacts the $NO_3^-$, $NH_4^+$ and H$_2$O **(aerosol water content)** concentrations as shown in Fig. 3, which is a consequence of gas-aerosol partitioning of the species."

Section 4.2 last paragraph: It would be useful to compare the uncertainties from the non-linearities to the model uncertainties that can be drawn from the comparison with the observations. This comparison has been done in Part 1 but this information has not been given in the present paper.
A new section "4.1 Evaluation of the predicted concentrations" has been added. It summarizes the main finding of the evaluation done in the Part 1.
There is also a new figure S2 showing timeseries of PM$_{10}$ concentrations measured by AirBase stations and compared to the EMEP model predictions. These examples show a set of cities for different dates.

It is also important to know that an evaluation of the meteorological fields has been performed and a Table S2 is provided in the supplement. This summarizes this evaluation over the 34 cities during the episode.

The following sentences have been added in the new section 4.1:
"The differences seen with the measurements may also be related to uncertainties in the regional emission inventory as regards to local situations and in the meteorological fields since forecasted meteorological fields have been used, but the impacts of the choice of the emission inventory and of the meteorological fields have not been addressed in this work. However, the meteorological conditions as used in the EMEP model were well represented over most of the cities, as shown in the comparison with the measurements of the NOAA Integrated Surface Hourly Data Base (https://www.ncdc.noaa.gov/isd) in Table S2. For example, by gathering all cities, the wind speed at 10 m has a correlation coefficient of 0.84 and a normalised mean bias of 8.08%, the relative humidity at 2 m has a correlation coefficient of 0.59 and a normalised mean bias of -2.38%, and the temperature at 2 m has a correlation coefficient of 0.95 and a normalised mean bias of -0.13%. It is worth noting in some cities, the wind speed is overestimated, which may cause an overestimation in the dispersion of the pollutants."

Section 5.2: The choice of the two cities (Paris and London) needs to be justified at the beginning of this section. Also, it would be useful to have the emissions for these cities for a more thorough interpretation of the results.

The idea was to compare the $PM_{10}$ over two major European capitals which experienced different influence (city vs country) and different composition (primary vs secondary components).

Please note, there is this small correction in the text, highlighted in bold:

"As illustration of the episode, **a focus on the two large European cities has been decided**, Paris and London".

There is also a new table S3 which summarizes the emission for both cities.

The following sentences have been added:

"It is worth noting that Paris had larger PM (fine and coarse), and SOx emissions during this period than London as shown in Table S3. At the opposite, London was characterized by larger CO and NH3 emissions."

Lines 316-318: The author's analysis is consistent with an air quality report for Paris. Could the author explain what is the information in this report that supports his analysis?

The following sentence has been changed:

"A report from the Paris regional air observatory (see https://www.airparif.asso.fr/_pdf/publications/pollution-episode-paris-area_dec2016.pdf) concluded the large $PM_{10}$ concentrations were mainly related to local sources such as wood burning and traffic."

Line 321: Reference to policies. The author may be more careful regarding policies since the present study assumes that the reduction of emissions applies to all sectors and with the same magnitude. In reality, policies on emissions during pollution events cannot be applied to all sectors and with the same level of regulation. The conclusion section gives a summary of the methodology and results but little discussion on the other sources of uncertainties of the method than the non-linearity (for instance the meteorology or the parameterization of wet deposition) and also very few prospects of extension of the methodology (for instance to PM2.5 or to the contribution of anthropogenic emissions by sector).

Some corrections have been done.

That's right about the policies, so the sentence has been changed as below:

"This domination of the primary components for this case also shows if the local emissions were reduced over this area during the 02 December, the level of urban background $PM_{10}$ could have been below the daily 50 ug/m$^3$ as recommended by WHO."

Note, the following sentence has also been added in the new Section 4.1 "Evaluation of the predicted concentrations":

"The differences seen with the measurements may also be related to uncertainties in the regional emission inventory as regards to local situations and in the meteorological fields since forecasted meteorological fields have been used, but the impacts of the choice of the emission inventory and of the meteorological fields have not been addressed in this work."

And in the conclusion:

"Other sources of uncertainties, such as the meteorological fields used for these predictions have not been addressed in this work. It is worth noting a good agreement has been found with meteorological observations over most of the cities."

About the extension of the work, a new sentence has also been added in the conclusion:

"Moreover, details on the sectoral contribution, which is not provided in this work, should be an important information to further describe this episode."

**Technical corrections**

Line 95: replace 'few' by 'a few'.
Done

Line 148: Replace 'Precipitations are' by 'Precipitation is'.
It has been replaced.

Line 149: Replace 'precipitations occur' by 'precipitation occurs.
Changed.

Line 165: Replace 'data set' by 'dataset'.
Done

Figure1 is small and it is very difficult to see the domains. More generally, most figures are small and uneasy to read.
The figures have been enlarged.

Line 215: Replace 'was developed' by 'occurred'.
It has been replaced.

Line 299: Replace 'larger' by 'largest'.
It has been changed.

Line 309: 'the list of country contributors are' to be replaced by 'the list of country contributors is'.
It has been corrected.

Line 310: Change 'correspond' to 'corresponds'.
Changed.

Line 338: Change 'larger' by 'a larger'.
Done

**Reviewer 2**

This study estimated the source contribution of PM10 concentrations based on a regional air quality forecasting model, and on a scenario approach in European cities. It was found that 20% of the predicted PM10 are from the city contributions (composed of primary PM components) and 60% are from the countries of the regional domain (excluding city contribution), and rest are contributed from the natural sources.

The major weakness of this study is the lack of the model evaluation in terms of the meteorological condition, PM10 concentrations, and the PM10 components. I think this information are strongly needed to ensure that the emission source contributions simulated from the model is reliable. In particular, this information is very important for designing the emission reduction plan in the future. Without these information, I don't think any of the model results can be trusted.

I would like to start by thanking the reviewer 2 for his comments and questions. I hope I have clarified all the comments given by the reviewer and I am sure it helps to improve the manuscript. I have answered all the points by writing my reply in red.

Among the corrections, the revised manuscript provides now:
- more information on the evaluation of the predictions (on PM10 and meteorological fields)
- more details on the episode
- clarification on the chemical non-linearity
- information on the emissions
- a catalogue of the different contributions for each city and for each day.

1. Can author state why the focus of the air pollution problem is PM10 instead of PM2.5? It seems nitrate contributes a lot for PM10 in your study region; however, I assumed most nitrate were the fine particles.
I agree with the reviewer that $PM_{2.5}$ is one of the major problems in terms of air quality. However, I have focused on $PM_{10}$ since this paper is the second part of the study published in 2020 and focusing on $PM_{10}$. It is also important to note that the $PM_{10}$ was the first product available in this system (before to make the $PM_{2.5}$ product available online).

Yes, indeed, nitrate might be an important contributor, due to the car traffic. Agriculture with ammonia emission can also largely impact $PM_{10}$, even in winter. Generally, these two sources have a larger impact in spring.
In 2016, the EEA report 2018 shows the rural background nitrate concentrations ranged from 0.018 to 3.090 $\mu g/m^3$. For comparison sulphate was between 0.18 and 3.08 $\mu g/m^3$, elemental carbon between 0.075 and 2.319 $\mu g/m^3$ and organic carbon, between 0.44 and 14.71 $\mu g/m^3$.

Reference:
EEA Report No 12/2018, Air quality in Europe 2018, https://www.eea.europa.eu/publications/air-quality-in-europe-2018.

2. How serious is the PM10 problem in your study area? What is the characteristics of the PM10 in terms of seasonality and spatial variability? Is PM10 a serious air pollution problem in December?
The text has been changed as follow (see the corrections highlighted in bold):
"One of this pollutant, the particulate matter smaller than 10 µm ($PM_{10}$), is related to premature mortality at high exposure. The World Health Organization (WHO) has established a shortterm exposure PM$_{10}$ guideline value of 50 µg/m$^3$ daily mean that should not be exceeded in order to ensure healthy conditions (WHO, 2005). **The WHO has also established a stricter guideline value for the annual average at 20 μg/m$^3$. In Europe, even if the air quality has been improved during the last decade, 13% of the EU-28 urban population was exposed to PM$_{10}$ levels above the daily limit value and approximately 42 % was exposed to concentrations exceeding the annual WHO guideline value in 2016 (EEA report 2018). These PM$_{10}$ can be emitted locally or transported on long distance. Most of the episodes occur in winter (e.g. EMEP Status Report 1/2018). Indeed, in wintertime, these episodes are often caused by a combination of stagnant air conditions and enhanced use of wood burning for residential heating during cold weather situations. The agriculture and the road traffic have also a large impact even if these two sources are known to usually contribute to PM$_{10}$ pollution in spring (e.g. EEA report 2018; EMEP Status Report 1/2018).**"

3. There is lack of discussions of the emission distributions. A graphical demonstration of the emission distributions is very helpful for readers to have a good understanding of the PM10 problem in your study area.
The following sentence has been added in Section 3.1 and new figure is provided in a supplement:
"As mentioned in Section 2.1, these PPM are distinguished in the EMEP model for two size of aerosols, fine aerosols and coarse aerosols. Note that, except on NH$_3$, the main source regions of these anthropogenic emissions such as NO$_x$ and CO are located over the main urban areas as shown in Fig. S1."

4. There is a lack of the evaluation of the meteorological model performance and air quality model performance. Without this information, I don't think the model simulation results can be trusted.
This evaluation has been performed and a Table S2 is provided in the supplement. This summarizes the evaluation of the meteorological fields over the 34 cities during the episode.

The following sentences have been added in the new section "4.1 Evaluation of the predicted concentrations":
"The differences seen with the measurements may also be related to uncertainties in the regional emission inventory as regards to local situations and in the meteorological fields since forecasted meteorological fields have been used, but the impacts of the choice of the emission inventory and of the meteorological fields have not been addressed in this work. However, the meteorological conditions as used in the EMEP model were well represented over most of the cities, as shown in the comparison with the measurements of the NOAA Integrated Surface Hourly Data Base (https://www.ncdc.noaa.gov/isd) in Table S2. For example, by gathering all cities, the wind speed at 10 m has a correlation coefficient of 0.84 and a normalised mean bias of 8.08%, the relative humidity at 2 m has a correlation coefficient of 0.59 and a normalised mean bias of -2.38%, and the temperature at 2 m has a correlation coefficient of 0.95 and a normalised mean bias of -0.13%. It is worth noting in some cities, the wind speed is overestimated, which may cause an overestimation in the dispersion of the pollutants."
5. There is a lack of the PM10 composition comparisons between the model and observation. This information is strongly needed to demonstrate that the model result is reliable and can be used to discuss the emission source contributions.
I agree, an evaluation of the PM$_{10}$ composition will be relevant. However, no measurements of EC, POM, SOA, $NO_3^-$ $NH_4^+$ and $SO_4^{2-}$ were available in the AirBase data set used in the evaluation as done in the Part I for the selected cities.

In the current EEA portal (https://aqportal.discomap.eea.europa.eu/products/attainment-viewers/attainment-tables-v2/), these compounds are still not available.

A few stations measuring SIA and EC are available on the ebas portal (http://ebas.nilu.no/default.aspx). However, only a few EMEP stations which correspond to rural background stations are available on this portal and only daily means could be used. The number of points is very limited, and it is difficult to have a clear conclusion with these results as shown below:

[Figure]

Fig. Scatterplot of daily mean concentration of EC, NH4, NO3 and SO4 measured by EMEP stations and predicted by the EMEP model. All available days of observations between 01 and 09 December 2016 have been used and compared to each individual predicted day.

It might be possible the model overestimates $NO_3$, but four points at rural stations (2 daily means from 2 stations) is not sufficient to provide a real conclusion on $NO_3$ concentrations over European cities during the studied period.

6. An overview of the study episode in terms of the observation characteristics should be provided.
More details have been provided about this episode. Note a new figure S2 shows timeseries of $PM_{10}$ concentrations measured by AirBase stations and compared to the EMEP model predictions. These examples show a set of cities for different dates.

7. Line 165, the reference year of the anthropogenic emission data set is 2011. How representative is this old dataset when it is applied to discuss the current air quality conditions?
This is a good point. Before to answer this question, it is worth reminding that this air quality forecasting system aims to predict the background concentrations in cities.

The TNO-MACC III emission inventories used in this study, correspond to the inventory used during the beginning the development of this system.

The use of this inventory followed the requirement from the Copernicus program.
This inventory has also been used in the European Ensemble air quality system (e.g https://www.slideshare.net/CopernicusECMWF/how-are-regional-analyses-forecasts-and-reanalyses-produced-by-guidotti and https://atmosphere.copernicus.eu/regional-air-quality-production-systems). Thus, both systems have used a consistent inventory.

It is also worth noting the operational system is now using the CAMS-REG emission inventory (https://atmosphere.copernicus.eu/sites/default/files/2019-06/cams_emissions_general_document_apr2019_v7.pdf) which is the successor of TNO-MACC. This update has been done after I started this study and after I changed of employer.

8. Section 4.1, there is a lack of the discussions of the observation characteristics.
The evaluation of the $PM_{10}$ concentrations (predictions vs AirBase measurements) was provided in the part I. The initial idea was not to repeat the same work.
However, I agree this information might be missing in this part II. So, additional information has been provided in a new section 4.1.

Fig. 2, there are only model simulation results which is not sufficient to persuade that the source contribution is reliable. A comparison with observation data is needed (e.g. Comparison with observed PM10 and PM10 components, to ensure the reliability of the model results).
The comparison of the EMEP forecast for Paris as shown in Fig2 with measurements is now shown in the examples presented in Fig S2 as mentioned in the reply to the comment 6.
As mentioned in the reply to the comment 5, there was no sufficient observations of the PM composition to have a validation of these components.

9. Line 225, "the chemical reason of the non-linearity is revealed by the negative contributions to the predicted PM10 concentrations". Please clarify the sentence.
Now it reads:
"The sum of each contribution should correspond to the total $PM_{10}$ calculated by the reference run, but some differences can appear. By splitting the $PM_{10}$ concentrations for each contribution based on their sign, the negative $PM_{10}$ concentrations help to reveal the species impacted by the non-linearity and explaining the differences seen with the total $PM_{10}$ concentrations calculated by the reference run. On the other hand, the positive concentrations provide the information on the overall composition for each contribution."

10. Line 230, "The mean PM10 concentration in a smaller area is larger, showing that with a smaller grid, the PM10 is less diffused over the integrated area." I think the discussions are weird. Isn't the 1-grid cell the closest grid to the emission source so that it has the largest concentrations?
It is correct. To clarify this point, the sentence is now:
"The mean $PM_{10}$ concentration in a smaller area is larger, **since the 1 grid cell is the closest grid to the emission source and so the mean concentration is less dispersed than over a larger area**."

11. Line 232, "The rest of Europe PM10 is mainly influenced by nitrate". Here, the nitrate concentration should be in the fine particle mode. Please provide evidence that demonstrating the PM compositions are mainly composed by nitrate.
I am also curious why the nitrate occupy a large fraction of PM10 in Europe.
The section has been rewritten and additional figures in a supplement (Figs. S3 to S10) have been added. Please refer directly to the revised manuscript.

Among the corrections, there is this sentence:

"60% of the contributions to the surface $PM_{10}$ have been coming from the "Rest of Europe", essentially $NO_3^-$ (by ~35%). The two other secondary inorganic aerosols represent another important part of this "Rest of Europe" contribution, since the $SO_4^{2-}$ and $NH_4^+$ together represent almost 30%."

12. Line 249, why the nonlinearity only impact NO3, NH4 and H2O? What about SO2 and SO4?

It appears thanks to Fig. 3 that the non-linearity in $PM_{10}$ is mainly related to $NO_3$, $NH_4$ and $H_2O$.

$NO_3$ and $NH_4$ are driven by the thermodin equilibrium with $HNO_3$ and $NH_3$ which involved $H_2O$.

$SO_4$ is related to the $SO_2$ oxidation homogeneous by OH and also reactions in clouds with $H_2O_2$ and $O_3$ (e.g. Simpson et al., 2012, and https://wiki.met.no/_media/emep/EMEP_course_29-30apr2019_Aerosols.pdf).

By applying Eq 1 for each component, i.e. by comparing the sum of all contributors calculated for each $PM_{10}$ component and normalized by the component concentration, we can see the non-linearity mainly impacts $NO_3$, $NH_4$ and $H_2O$. $SO_4$ and SOA are also impacted by the chemical non-linearity, but it remains small. The calculation is done for all hourly concentrations (all 4-day forecasts) over all cities.

The following figure is focusing on the "Rest of Europe" contribution, which is the contribution presenting the larger non-linearity:

[Figure]

[Figure]

Note that the non-linearity on POM, natural, rest PPM and EC components is null.

13. Please explain why the eq 1 can be used to estimate the nonlinearity.

For each hourly contribution, the $PM_{10}$ calculated by the three perturbation runs (5%, 15% and 50% reduction) have been compared to the hourly $PM_{10}$ concentration calculated by the reference run.

The variance of these three estimates can be used to estimate the impact of the nonlinearity since in theory the three perturbated runs should provide the same hourly $PM_{10}$ concentration than the reference run.

The following information (in bold) has been added:

"**By comparing the three estimates from the perturbation runs to the total concentration for each contribution,** this gives an estimation of the impact of the non-linearity for each contribution. **In theory, the three perturbated runs should provide the same hourly $PM_{10}$ concentration than the reference run**."

This calculation is done for all cities and all 4-day forecasts.

14. Line 315, what is the source of the EC and PPM?

The forecasting system and the studies aim to provide the country (part 1) and city (part 2) contributions. There is no information on sectoral apportionment.

The residential combustion and transport (diesel combustion) are the most important sources for fine EC, while power plants and industry are the main sources for coarse EC in the emission inventory (Kuenen et al., 2014).

Rest Primary Particulate Matter (rest PPM) corresponds to the PPM excluding EC and OM (organic matter) (e.g. Tab S6 in Simpson et al., 2012; and description in Section 2.1).

However, if the reviewer is interested on the impact of different sectors on air quality (for other years), another tool is available on this link: https://policy.atmosphere.copernicus.eu/CAMS_ACT.php, based on simulations performed by the CHIMERE Chemistry-Transport Model. The complementary information on the country contribution in this tool is still calculated by the EMEP model.

Note this following sentence has been added in the conclusion:

"Moreover, details on the sectoral contribution, which is not provided in this work, should be an important information to further describe this episode."

15. Please provide evidence to support the SR result from model simulation.

As mentioned earlier, timeseries, comparing the EMEP $PM_{10}$ concentrations with airbase measurements over London and Paris have been added in Fig. S2.

Moreover, the analysis over Paris also refers to a report from Local authorities (see the reply to the comment 19).

To my knowledge, there is no study for both cities during this episode in terms of source apportionment.

16. It's not the reader's responsibility to read the PARTI of the companion study in order to understand this article. The author need to summarize the findings from the PARTI and explained in this study.

It is correct and I apologize for the missing information.

More information has been provided such as in the introduction, in the sections 3.1 and 4.2. A new section "4.1. Evaluation of the predicted concentrations" summarising results from the part 1 has also been added.

17. Use of "Local" contribution is very confusing.

The term "local" has been replaced through the manuscript and in figures by "city".

Note, the title has also been changed to be consistent with the Part 1.

Now it reads:

"Prediction of source contributions to urban background $PM_{10}$ concentrations in European cities: a case study for an episode in December 2016 using EMEP/MSC-W rv4.15 - Part.2 The city contribution".

18. Line 321, "if policies to reduce the local emissions over this area were performed during 02 December, the level of urban background PM10 would have been below the daily 50 ug/m3". This statement needs to be supported with more evidence because it involves with the policy decision.

Agreed. The sentence has been changed:

"This domination of the primary components for this case also shows if the local emissions were reduced over this area during the 02 December, the level of urban background $PM_{10}$ could have been below the daily 50 ug/$m^3$ as recommended by WHO."

19. Section 5.2, the discussions in Paris and London should be evaluated with the observed data to support the findings.

That's right. Some examples in the Fig S2, London and Paris are included.

Moreover, the local authorities (AirParif) confirmed the local origin of the episode described in this section. The following sentence has been changed:

"A report from the Paris regional air observatory (see https://www.airparif.asso.fr/_pdf/publications/pollution-episode-paris-area_dec2016.pdf) concluded the large $PM_{10}$ concentrations were mainly related to local sources such as wood burning and traffic.".

However, there is no comparable measurements on PM composition for this episode.

20. Line 351-352, the source-contribution is done based on a scenario approach; however, due to various experiments are needed to be conducted which take amounts of computational time. How can this be accomplished in an air quality operational mode?

For the forecasts published online, four-day meteorological forecasts (12 UTC forecast) from the IFS system of the ECMWF are retrieved daily around 18:15 UTC. Then, the EMEP simulations are run in parallel during the night.

The 12 UTC forecast from yesterday's forecast is used, so that there is sufficient time to run the EMEP forecasts well before the deadline for delivery at 08UTC. Only the results using a perturbation at 15% are used for the operational forecasts provided online. Thus, it reduces the number of runs compared to this study.

If available at the start of the forecast run, boundary conditions are taken from the C-IFS. If not, default BCs are specified for O3, CO, NO, NO2, CH4, HNO3, PAN, SO2, ISOP, C2H6, some VOCs, Sea salt, Saharan dust and SO4.

Note for this study, only the BCs from C-IFS were used.

21. What is the main objective of this study? To provide the source contributions for PM10, or to develop a near-real time system that provides the source contributions to PM10? I don't think the design of the current study meet the study objective. For example, the scenario experiment does not provide comprehensive information of the source contributions. The discussions of developing the real-time source contribution technique are not introduced.

I agree this work is not suited for mapping local exceedances such as street canyons and industrial sites, but as already mentioned in the introduction, the system uses a relatively coarse resolution to define the cities.

Thus, the aim of this study is to describe the near-real time system for assessing $PM_{10}$ background concentrations in the urban area.

The following sentences have been changed in the introduction:

"Thus, by combining the information from the country contribution given in the companion paper and the city contribution presented hereafter, the system allows providing information on long-range transport in the European cities and the pollution coming from the urban area. These contributions might be important to determine short term air pollution control measures, which can remain difficult to assess by local authorities."

And also:

"Thus, the objective of this study is to present the near-real time calculation of the urban background contribution predicted by the EMEP/MSC-W model on hourly resolution for each capital of the 28 European Union countries plus Barcelona, Bern, Oslo, Reykjavik, Rotterdam and Zurich".

In addition, the term "local" has been replaced in the manuscript by "city".

**Reviewer 3**

Dear reviewer 3, I would like to thank you for your careful reading of both manuscripts: this paper (part 2) and the companion paper (part 1). I really appreciate your useful comments which help to improve the manuscript.
You will find my answers to your several points, written in green.

For editor
The author changed the definition of "Local" from country to city and produced another paper. There is nothing innovative in the current study but giving some valuable information that PM10 in most cities in Europe is mainly attributed to the area in domestic country in addition to the city. In addition, the author can provide information like "the contribution of "local", "domestic country in rest of Europe", "other countries in rest of Europe", and "Extra", in terms of each city. Therefore, the reviewer think it is worth to be read widely. However, there are some points the reviewer cares, e.g., why the author used forecast meteorology instead of retrospective? the method of nonlinearity calculation, etc. Therefore, the reviewer suggests the outcome of this review is "major revision". The reviewer is willing to review the revised manuscript for the next submission.
Even if this comment was addressed to the Editor, I would like to add additional comments. That's right, this work is the following of the previous manuscript (part 1). I thought I made it clear within the summary and in the whole manuscript, so I apologise if it was unclear.
The part 1 was focusing on a PM episode on Dec 2016. The aim of this first paper was to compare the response of two models, using two techniques to calculate the country contribution over European cities. This was the first study related to the development of this forecasting system. This part 1 demonstrates the ability of two modelling approaches to identify source contributions of particulate matter from different countries to several cities in Europe during a pollution episode. The results showed a large degree of similarity which is a key result.
However, only one model (EMEP/MSC-W) provides an additional information in these forecasts. The EMEP/MSC-W model also calculates the local urban background (in addition to the country contribution). This information was not provided in the Part 1. This is another important information and it was beyond the scope of the first paper. It is the reason of the writing of this second study,
In addition to the several references related to this first study, the second paper was already mentioned in the part 1 manuscript (last sentence in the conclusion).

The answers to the questions on the meteorological fields used and on the provision of the contributions over all cities are given in the rest of this document.

For authors
General comments:
The reviewer used to think that the chemical non-linearity is the chemical reaction between sources. In this study, the authors used the ratio of standard deviation of hourly concentration to hourly concentration. What is the principle or base for their method? Why the nonlinearity is calculated based on statistics instead of chemistry? In addition, the authors cited Pommier et al. (2000) a lot. It is ok to cite a companion paper but the reader is not obligated to read the companion paper. Therefore, some information should be explained or mentioned in the current manuscript. At last, the reviewer thinks although the current study is not innovative compared with the Part I study but still provide a valuable information: "the contribution of "local", "domestic country in rest of Europe", "other countries in rest of Europe", and "Extra".

Therefore, the reviewer suggests the author to list a table to provide such information in terms of every city.

I agree with the reviewer about the chemical non-linearity since it was mentioned in the manuscript. However, I have made some corrections to clarify this point (e.g Sections 3.2, 4.3)

In addition to these corrections:

- In Section 2.1, in the description of the model, there is now this sentence:
"The EMEP model uses the MARS equilibrium module of Binkowski and Shankar (1995) to calculate the partitioning between gas and fine-mode aerosol phase in the system of $SO_4^{2-}$-HNO$_3$-$NO_3^-$-NH$_3$-$NH_4^+$. This module also calculates the mass of aerosol water (Simpson et al., 2012). This calculated mass of water is added to both dry PM$_{10}$ masses when being compared with measured concentrations."

- In section 4.2., there is this sentence:
"This impacts the $NO_3^-$, $NH_4^+$ and H$_2$O (aerosol water content) concentrations as shown in Fig. 3, which is a consequence of gas-aerosol partitioning of the species."

- About the complementary information from the Part 1, additional information through the manuscript has been provided such as this paragraph in Section 3.1:
"Thus, all these simulations are a complementary information of the country contributions presented in Pommier al. (2020). Indeed, in the country contribution calculations provided in Pommier et al. (2020), there is the "Domestic country" which represents the country corresponding to the studied city (e.g. Spain for Barcelona). Another contributor in the country SC is "30 European countries". In the country SC, the contributions for 31 countries are calculated which include the 28 EU countries, Iceland, Norway and Switzerland, and this "30 European countries" combines all these contributors and excludes the "Domestic country"."

- A new Section 4.1 "Evaluation of the predicted concentrations" is provided and Section 4.2 has also been rewritten with more results.

- A catalogue (csv file) of the different contributions for each city and for each day has been provided.

Special comments:
1. Please the model evaluation of meteorology, PM10, and PM10 compositions before any discussion in the manuscript. Readers are not obligated to read the Part I manuscript. Thus the authors should narrate or at least mention the model performance clearly.
In the revised manuscript, the reviewer will find a new section "4.1 Evaluation of the predicted concentrations". It includes an evaluation of the PM$_{10}$ concentrations and also an evaluation of the meteorological fields. It is also worth noting in the following section "4.2 Origin of the PM$_{10}$", there is more information on the meteorological conditions during the episode.

I also agree, an evaluation of the PM$_{10}$ composition will be relevant. However, no measurements of EC, POM, SOA, $NO_3^-$ $NH_4^+$ and $SO_4^{2-}$ were available in the AirBase data set used in the evaluation as done in the Part I for the selected cities.
In the current EEA portal (https://aqportal.discomap.eea.europa.eu/products/attainment-viewers/attainment-tables-v2/), these compounds are still not available.

A few stations measuring SIA and EC are available on the ebas portal (http://ebas.nilu.no/default.aspx). However, only a few EMEP stations which correspond to rural background stations are available on this portal and only daily means could be used. The number of points is very limited, and it is difficult to have a clear conclusion with these results as shown below:

[Figure]

Fig. Scatterplot of daily mean concentration of EC, NH4, NO3 and SO4 measured by EMEP stations and predicted by the EMEP model. All available days of observations between 01 and 09 December have been used and compared to each individual predicted day.

It might be possible the model overestimates $NO_3$, but four points at rural stations (2 daily means from 2 stations) is not sufficient to provide a real conclusion on $NO_3$ concentrations over European cities during the studied period.

2. On line 71, a comma or no blank between 400 and 000 is suggested.
It has been corrected.

3. On line 115, please explain the "concept" clearly. On line 116, please explain the meaning of "coarse".
On line 116, the following changes have been made:
"It is worth noting, the definition uses a relatively coarse resolution (at least 0.25° longitude × 0.125° latitude) which is representative of the background concentration, and is comparable to the definition of the city domain used in previous studies such as in Thunis et al. (2016) who used an area of $35 \times 35$ km$^2$ or in Skyllakou et al. (2014) who used a radius of 50 km from the city center. Thus, 1 model grid cell (0.25° longitude × 0.125° latitude), 9 grid cells and the grid cells covering the definition given by the Global Administrative Area - GADM) have been used as also done in Pommier al. (2020)".

Moreover, the sentence having the word "concept" has been deleted and this paragraph has been rewritten.

4. Section 2.1, EMEP is not a meteorology-chemistry coupled model. Please supplement the description of meteorological inputs in current manuscript.

These new sentences and Table S1 have been added:

"Meteorological data are normally required at 3-hourly intervals for the EMEP model. The EMEP model has systems for deriving parameters when missing or can do without some meteorological fields such as the 3D precipitation explained above. Table S1 summarises the meteorological fields used in the EMEP model. Vertically, the fields are interpolated onto the 20 EMEP σ levels."

**Table S1.** Input meteorological data used in the EMEP Model.

| Parameter | Unit | Description |
|---|---|---|
| **3D fields** | | **for $\sigma$ levels** |
| u,v | $m/s$ | Horizontal wind velocity components |
| q | $kg/kg$ | Specific humidity |
| $\theta$ | $K$ | Potential temperature |
| CW | $kg/kg$ | Cloud water |
| CL | $\%$ | 3D Cloud cover |
| cnvuf | $kg/sm^2$ | Convective updraft flux |
| cnvdf | $kg/sm^2$ | Convective downdraft flux |
| PR | $mm$ | Precipitation |
| **2D fields** | | **for surface** |
| PS | $hPa$ | Surface pressure |
| T2 | $K$ | Temperature at $2m$ height |
| RH2 | $\%$ | Relative humidity at $2m$ height |
| SR | $W/m^2$ | Surface flux of sensible heat |
| $\tau$ | $N/m^2$ | Surface stress |
| SST | $K$ | Sea Surface Temperature |
| SWC | $m^3/m^3$ | Soil water content |
| lspr | $m$ | Large scale precipitation |
| cpr | $m$ | Convective precipitation |
| sdepth | $m$ | Snow depth |
| ice | $\%$ | Fraction of ice |
| SMI1 | | Soil moisture index level 1 |
| SMI3 | | Soil moisture index level 3 |
| u10, v10 | $m/s$ | Wind at $10m$ height |

Please note, if the reviewer would like to have more information on the model, details are provided in the user guide: https://github.com/metno/emep-ctm/releases/tag/rv4_15
There are also more details in Section 3 in Simpson et al. (2012).

5. Section 2.2, is this study a forecast run or a retrospective run? Please narrate clearly. For such kind of study, a retrospective run is better than forecast run since the meteorology is the reanalysis data and closer to observations.

That's right but as explained in the manuscript, the forecasted meteorological fields were used to be consistent with the part I. Moreover, these meteorological fields are the inputs used in the operational mode. Thus, this work and the companion paper are a clear illustration and evaluation of this forecasting system.

For this reason, the following sentence was used in the current manuscript:

"These forecasted meteorological fields correspond to the fields which were used in the online production for these dates and used in the companion paper (Pommier et al., 2020)."

6. On line 166, please check the URL. The reviewer could not find the webpage.
The link has been changed.

7. On line 188, the choice of 15% is just because it is large enough to show clear concentration changes? Is there a stronger reason? Moreover, non-linearity represented less than 2% of total concentrations for each predicted country contributions but may be larger for cities. Please reconsider your narratives.
The 15% reduction has been chosen since it is significant enough to provide concentration changes and it avoids a large impact of the chemical non-linearity such as by using a 50% reduction.
The sentence has been changed as follow:
"In the companion paper, it was shown that the non-linearity, related to the emissions reduction used, represented less than 2% of the total concentrations over each city (Pommier et al., 2020)."

8. The authors used zero-out emissions of two cities as a run. Is there any test that has proved hardly interaction exists between these two cities?
Yes, a test has been performed by using the 15% perturbation.
The perturbated simulations has been done for all cities individually, for the three city definitions and compared to the simulations using the pair of cities.

The following figure shows the mean difference for $PM_{10}$ and its components between the simulations using the perturbation on the pair of cities and the simulations using the perturbation on each individual city.
This average is done for all cities and all forecast hourly concentrations, gathered by contribution. The larger difference is calculated by using the 9 grid cells definition, but it remains negligible.
The vertical bars represent the standard deviation.

[Figure]

9. On line 193, please explain the method of perturbation run clearly.

Now it reads:

"The perturbation runs have been performed for each capital of the 28 European Union countries plus Barcelona, Bern, Oslo, Reykjavik, Rotterdam and Zurich. These simulations over these selected cities, in comparison with the reference run, give the contribution for each city."

10. On line 197, does the "Rest of Europe" include the domestic country? In other words, e.g., all areas in Europe in addition to Paris, right?

"Rest of Europe" corresponds to all countries within the regional domain, but excluding the studied city. Thus, for Paris, France is included in "Rest of Europe", France being the "domestic country" in the country SC (Part 1). Paris itself is the "City" contribution.

11. On line 198, "Then, this "Rest of Europe" contribution……..by the difference with the "Local" contribution" is suggested to "Then……..by the difference between the total and "Local contribution"".

Actually no.

The Rest of Europe contribution is related to the run where the anthropogenic emissions of all the countries in the regional domain are perturbated. Then, we estimate this contribution by subtracting the contribution from the city.

To clarify this point, the sentence has been changed as below:

"Since this additional perturbated run also includes the cities, this "Rest of Europe" contribution has been calculated by subtracting the "City" contribution."

However, the explanation of the "Extra sources" contribution was also missing, and it is also linked to the comment 24. The following sentence has been added:

"The remaining $PM_{10}$ which are neither included in the "City" contribution nor in the "Rest of Europe" contribution are listed in the "Extra sources" contribution which is mainly represented by the BCs and natural sources (sea salt, forest fires and dust)."

12. On line 201, please narrate "scaled by 15%" more clearly.

The following sentences have been added:

"By differentiating over the studied area, the concentration from the perturbed run with the concentration provided by the reference run, we have an estimation of the influence of the source (i.e. city). By scaling with the reduction used, it gives the estimated concentration related to the source."

13. On line 209, the reviewer could not understand the meaning of 9 "dates"? Is the simulation executed daily? Besides, "9 rest of EU", why?. What is the 9 reference runs?

I am sorry if it was unclear. There are 9 reference runs and 9 rest of Europe runs, since there is one run per day (from 01 to 09 December) for both scenarios. As mentioned in Section 2.2., "the predicted fields have been used to initialise successive four-day forecasts".

This is a forecast so every day there is a 4-day forecast for each pair of city, the reference run and the rest of Europe run.

It has been changed and now it reads:

"By using these three different perturbations, the total number of simulations performed for this study is equal to 495: 17 pairs city × 9 dates (from 01 to 09 Dec) × 3 perturbations (5%, 15%, 50%) + 9 rest of Europe (one per day) × 3 perturbations (5%, 15%, 50%) + 9 reference runs (one per day)."

14. On line 220, please give a strong reason why the reader is suggested to compare the Fig. 2 in current manuscript and Fig. 1 in in Pommier et al. (2020).

The sentence is now:

"As a complementary information, the reader is invited to compare with the Figure 1 in the companion paper, presenting the country contributors for the same time-series. By combining the information from both timeseries, it is clear that the contribution from France in Paris was largely influenced by the city itself and not only by the rest of the country."

15. On line 223, please explain "4-d" predictions.

It has been corrected. Now it reads "4-day".

16. On line 239, what are the other sources (30-40%) for "Extra sources"? Are they the BCs?

As explained in lines 219-220, "Extra sources" represent mainly BCs and natural sources. However, there is also a lower contribution from ship traffic, biogenic sources, aircraft emissions, and lightning.

17. Is the variance between city to city and date to date large? Is it proper to express in mean concentrations?

This sentence has been deleted.

The section has been rewritten and additional figures in a supplement (Figs. S3 to S10) have been added. Please refer directly to the revised manuscript.

18. On line 241, please calculated the proportion of the "local", "Domestic country" in "Rest of Europe", "Rest of Europe" not including the "Domestic country", for example, Paris.

The contribution of the "local" (city) contribution is already calculated and shown in Fig.6. However, I have put it hereafter. You can see below the different contributions in the following figures (City, Rest of Europe and External sources) for each individual day and for all cities. The results are for the 9 grid cells and using the 15% perturbation.

[Figure]

[Figure]

However, I do not understand the difference in the question between, "Domestic country" in "Rest of Europe", "Rest of Europe" not including the "Domestic country".
In the following figures, I have plotted:
- "Domestic country",
- "Domestic country" – "City" and
- "Rest of Europe" (+ "City") - "Domestic country" contributions. City has been added into Rest of Europe since it is included in the Domestic country

Thus, by subtracting two contributions, e.g. "Domestic country" – "City", if the value is large, it shows the second contribution (city in this example) is low compared to the first contribution (Domestic country in this example).
The results over Paris, as requested, are given in the table below.

[Figure]

9 grids - pertubation factor: 15%

9 grids - pertubation factor: 15%

[Figure]

For Paris, the mean contribution in percent for each date was:

| Contribution | 1 Dec | 2 Dec | 3 Dec | 4 Dec | 5 Dec | 6 Dec | 7 Dec | 8 Dec | 9 Dec |
|---|---|---|---|---|---|---|---|---|---|
| Local (city) | 44 | 28 | 14 | 16 | 32 | 29 | 24 | 25 | 23 |
| Rest EU | 39 | 57 | 51 | 73 | 65 | 65 | 69 | 82 | 83 |
| External | 17 | 15 | 35 | 11 | 3 | 6 | 7 | -6 | -6 |
| TOTAL = 100% | | | | | | | | | |
| | | | | | | | | | |
| A low value means the city contribution is large in the Domestic country contribution: | | | | | | | | | |
| Domestic country* – Local (city) | 29 | 27 | 14 | 26 | 43 | 52 | 56 | 61 | 54 |
| A low value means the Domestic country contribution is large in the Rest of Europe contribution: | | | | | | | | | |
| Rest EU – Domestic country* ++ | 10 | 30 | 37 | 47 | 22 | 13 | 13 | 21 | 29 |

*Domestic country is provided by the Country Source Contribution presented in the Part 1.*
*++ Note for this calculation, the city contribution has been added to Rest of Europe, since it is included in the Domestic country.*

To explain these results, I give the example on 01 December.
At this date, 44% of the $PM_{10}$ was from the city, 39% from the rest of Europe, which includes France (Domestic country) and the countries in the regional domain, and 17% from external sources (mainly BCs and natural).

By completing these source contributions, with those calculated in the Part 1, we have:
Only 29% was from the Domestic country (France) by excluding the city (Paris). Only 10% was from the Rest of Europe, excluding the Domestic country (France) and the city (Paris).

The effect of the non-linearity is highlighted by the negative contribution on 8 and 9 December.

Another way to present the contribution, and now included in the paper as a new Figure 7, is to calculate the ratio between two contributions as below:

[Figure]

**Figure 7: Mean ratio of Domestic Country contribution (excluding the City contribution) to the Rest of Europe contribution in percent, for each city from 01 to 09 December 2016. Each city edge is defined by 9 grid cells. The contribution is based on the calculations performed by the 15% perturbation runs. The City contribution has been removed from Domestic Country contribution since it is not included into the Rest of Europe contribution.**

This ratio is also presented in the catalogue provided in the supplement.

With the following text in Section 5.1
"Figure 7 shows this large impact of the "Domestic country" in the "Rest of Europe" contribution in most of the cities, except on the Central European cities and in Benelux impacted by the surrounding countries. Note that cities such as Nicosia and Valetta were mainly influenced by the "Extra sources" contribution which was essentially related to natural sources and BCs."
And
"A catalogue summarizing the mean of these hourly contributions for each individual day has been provided in the supplement. The three contributions (City, Rest of Europe and Extra Sources) are presented as well as the Domestic country contribution. The catalogue also provides the information on the mean part of City in the Domestic Country contribution, the mean part of the Domestic Country in the Rest of Europe contribution and the $PM_{10}$ daily mean concentration."

19. On line 246, the chemically non-linear effect is negative. Please denote the negative term is which minus which. On line 251, if NH4NO3 is formed by NOx and NH3 in different regions, there is additional PM10 formed. Therefore, the non-linear effect is positive, isn't it?
Yes, a positive non-linear impact is possible.
To better explain the idea to split the concentration based on the sign, the following sentences have been added:
"The sum of each contribution should correspond to the total $PM_{10}$ calculated by the reference run, but some differences can appear. By splitting the $PM_{10}$ concentrations for each contribution based on their sign, the negative $PM_{10}$ concentrations help to reveal the species impacted by the non-linearity and explaining the differences seen with the total $PM_{10}$ concentrations calculated by the reference run. On the other hand, the positive concentrations provide the information on the overall composition for each contribution."

20. On line 256, "If this NOx is emitted in excess", why do the authors use "If" in this sentence?
It has been changed as hereafter (in bold):
"**When** $NO_x$ is emitted in excess, i.e. within a $NH_3$ limited regime, a $NO_x$ emission reduction will have a small effect at the receptor point."

21. On line 262, "it is very small", what is "it", "the impact of the percentage" or "the size of the city edges"?
It has been changed by "… the impact of both parameters is very small…".

22. On line 271, Please explain the formula (1) is reasonable and persuasive.
For each hourly contribution, the $PM_{10}$ calculated by the three perturbation runs (5%, 15% and 50% reduction) have been compared to the hourly $PM_{10}$ concentration calculated by the reference run.
The variance of these three estimates can be used to estimate the impact of the nonlinearity since theoretically the concentrations from these three estimates should be equal to the concentrations given by the reference run.
The following information (in bold) has been added:
"**By comparing the three estimates from the perturbation runs to the total concentration for each contribution**, this gives an estimation of the impact of the non-linearity for each contribution. **In theory, the three perturbated runs should provide the same hourly $PM_{10}$ concentration than the reference run**."
This calculation is done for all 34 cities and all 4-day forecast (9 dates × 96 predicted concentrations).

23. On line 271, n=3, is that representing standard deviation reliable in the view of statistics?
Indeed, three emission reductions have been tested in this work (5%, 15%, 50%). It is correct by thinking more perturbations will help to build a more statically significant data set.
However, the reader must remind to perform this work, 495 simulations was already performed. By adding an additional perturbation will add 153 simulations (17 pairs of cities × 9 studied dates).
This number of simulations does not take into account the postprocessing of the data.
The number of hourly concentrations for the "city" contribution which has been calculated is 88,128 (34 cities × 9 dates × 96 predicted concentrations × 3 perturbations), compared to the 29,376 reference concentrations for all cities (34 cities × 9 dates × 96 predicted concentrations).

24. On line 275, "It is worth noting ……other contributions". Please explain clearly.

This "extra sources" contribution is calculated by subtracting the total $PM_{10}$ to the "city" and the "Rest of Europe" contributions.
Thus, the non-linearity in these two contributions will influence the non-linearity in the "extra sources" contribution.

The information on the calculation on the "extra sources" contribution was missing, see the answer to the question 11.

The sentence in line 275 has been changed as below:
"It is worth reminding the "Extra sources" contribution is calculated by subtracting the total $PM_{10}$ concentrations to the two other contributions. Thus, the non-linearity from the "Extra sources" depends on the non-linearity of the two other contributions."
A good example is the response to the question 18 with the negative contribution in "Extra sources" over Paris.

25. On line 277, "The limited impact of …..are robust". Please explain clearly.
 Now it reads (see changes in bold):
"The limited impact of the non-linearity in the mean values, **highlighted by the small values in Figure 4**, shows that the responses to perturbation runs are robust. **Indeed, this shows the sum of all contributions is equivalent to total $PM_{10}$ concentration**".

26. On line 298, the superscript is not needed for dates.
It was done automatically with word. It has been changed.

27. On line 300, the underestimated hourly PM2.5 doesn't mean the "Local" PM2.5 is also underestimated. The underestimation could be due to other sources.
It's right. It has been corrected (see in bold):
"It is possible that the fraction of "**city**" $PM_{10}$ is underestimated, **as the other contributions**, by the model."

28. Fig. 7, Fig. 8, please denote the full name of countries. Not everyone understands the abbreviations of countries.
It has been added in the caption of Figs 7 and 8.

29. Fig. 7 captions, what is "countries not included in the country SC runs"? Please explain it clearly in the current manuscript.
This sentence has been added in Section 5.2:
"The other countries in the regional domain but not used in the country SC are gathered in the "External" contribution with the BCs."

Note the calculations done in the country SC are explained in Section 3.1:
"In the country SC, the contributions for 31 countries are calculated which include the 28 EU countries, Iceland, Norway and Switzerland, and this "30 European countries" combines all these contributors and excludes the "Domestic country"."

30. Fig. 7 captions, "The five main contributors are plotted as well as the difference between the daily mean and the sum of these five contributors (Rest).". This sentence should be split to two sentence: "The five main contributors are plotted as well." and "The "Rest is the difference between the daily mean and the sum of these five contributors", right?

It has been changed. Now it reads: "The five main contributors are plotted. The "Rest" is the difference between the daily

---

## Author Response (AR2)

Comments to the Author:
Dear Authors,

I have now received two reviewers' comments on your manuscript. Please address them thoughtfully (especially R2's Q5), and let me know if you have any questions.

Dear Min-Hui, dear reviewers.
Please find hereafter my replies to your additional comments. Note that in the revised manuscript, the previous corrections are highlighted in red and the new corrections in blue.

R1:
1. Section 2.1, spelling of DMS is incorrect.
It has been corrected.

2. Section 2.2, author used 2011 emission data to simulate 2016 case. Section 4.1 mentioned that the comparison between model and observation can be due to emission uncertainty. But this study does not include any discussions related to the emission uncertainty.
The following information **(in bold)** has been added:
"The TNO-MACC emission dataset for 2011 on 0.25° × 0.125° (longitude-latitude) resolution (Kuenen et al., 2014, see https://atmosphere.copernicus.eu/sites/default/files/repository/MACCIII_FinalReport.pdf) has been used and the forest fire emissions are from GFASv1.2 inventory (Kaiser et al., 2012) **as done in the companion paper and at the beginning of the development of the product. It is worth noting the use of a more recent CAMS emission product (CAMS-REG, Granier et al., 2019) has not been addressed in this work.**

With the corresponding reference:

Granier, C., S. Darras, H. Denier van der Gon, J. Doubalova, N. Elguindi, B. Galle, M. Gauss, M. Guevara, J.-P. Jalkanen, J. Kuenen, C. Liousse, B. Quack, D. Simpson, K. Sindelarova: The Copernicus Atmosphere Monitoring Service global and regional emissions (April 2019 version) Report April 2019 version, doi:10.24380/d0bn-kx16, 2019.

3.In Fig. 3, why can use of large city areas prevent non-linearity? Considering the emission characteristics, are the PM compositions reasonable? close to the real world observation?
A smaller area will be more influenced by external sources than a large area (such as 9 grid cells definition). Thus, it will increase the impact of non-linearity in the calculated sources.
It is also worth reminding that the forecasting system aims providing information on background concentration, so the emission characteristics and the composition are reasonable even in comparison with "real world" observations.
This system does not obviously provide concentrations and contributions at street-level.

4.Section 5.1. For Paris, the largest peaks are predicted on December 01st and on 02nd (e.g Fig. 2). On December 1st, the "City" contribution represented in average 44% of the PM10. Figure 2 only presents variation from Dec 2.
I have added the following information (in bold) in the sentence:
"On December 1st, the "City" contribution represented in average 44% of the $PM_{10}$ **(see catalogue)."**

5. This study used an old scenario to demonstrate the impact. Authors need to include discussions about the more recent update regarding the system development, such as emission inventory, etc, in the manuscript.

Note there is a new information on the emission inventory, as mentioned in your comment (2).
In addition, the following sentence has been added in the conclusion:
"The use of more recent emission inventories such as CAMS-REG has also not been studied in this work."

R2:
1. The remark on the basis for the choice of the precipitation intensity has not been fully addressed. Please complete the answer and manuscript.
The following information (in bold) has been added:
"The intensity of the precipitation is assumed constant over all heights where they are non-zero **and is set equal to surface precipitation intensity**."

2. A short discussion on the choice of square shape for the areas (as in the response to referees) should appear in the revised paper.
The following sentence has been added in Section 2.2.:
"The advantage to have a city domain defined by the 1 grid cell or 9 grid cells, is to have a similar domain for all cities used for the comparison. By using the grid cells based on GADM definition, the size of the cities differs according to the administrative extension of each city."